# Nanoscale cellular organization of viral RNA and proteins in SARS-CoV-2 replication organelles

Leonid Andronov [1,7], Mengting Han [2,7], Yanyu Zhu [2], Ashwin Balaji[1,3], Anish R. Roy [1], Andrew E. S. Barentine[1], Puja Patel[4], Jaishree Garhyan[4], Lei S. Qi[2,5,6] ✉ & W. E. Moerner [1,5] ✉

The SARS-CoV-2 viral infection transforms host cells and produces special organelles in many ways, and we focus on the replication organelles, the sites of replication of viral genomic RNA (vgRNA). To date, the precise cellular localization of key RNA molecules and replication intermediates has been elusive in electron microscopy studies. We use super-resolution fluorescence microscopy and specific labeling to reveal the nanoscopic organization of replication organelles that contain numerous vgRNA molecules along with the replication enzymes and clusters of viral double-stranded RNA (dsRNA). We show that the replication organelles are organized differently at early and late stages of infection. Surprisingly, vgRNA accumulates into distinct globular clusters in the cytoplasmic perinuclear region, which grow and accommodate more vgRNA molecules as infection time increases. The localization of endoplasmic reticulum (ER) markers and nsp3 (a component of the double-membrane vesicle, DMV) at the periphery of the vgRNA clusters suggests that replication organelles are encapsulated into DMVs, which have membranes derived from the host ER. These organelles merge into larger vesicle packets as infection advances. Precise co-imaging of the nanoscale cellular organization of vgRNA, dsRNA, and viral proteins in replication organelles of SARS-CoV-2 may inform therapeutic approaches that target viral replication and associated processes.

Due to its global health impact, the SARS-CoV-2 betacoronavirus and its infection of mammalian cells have been the subject of a large number of studies across multiple fields. Biochemical methods have allowed researchers to investigate the interactions between the viral RNA and the host proteins in vitro and in cellular extracts, leading to much insight[1,2]. There have also been electron microscopy (EM) studies of resin-embedded samples as well as vitrified samples using cryo-electron tomography, all of which have been profiting from the large increase in EM resolution and contrast in recent years. These EM studies can provide very high-resolution structures of protein complexes as well as tomograms of organelles in the cellular context. High-contrast filamentous structures and membranes appear regularly in such images, allowing identification of single- and double-membrane vesicles (DMVs)[3–5]. However, the all-important viral RNA and associated proteins are challenging to identify by EM due to a lack of specific contrast. While some researchers have detected RNA-like

[1]Department of Chemistry, Stanford University, Stanford, CA 94305, USA. [2]Department of Bioengineering, Stanford University, Stanford, CA 94305, USA. [3]Biophysics PhD Program; Stanford University, Stanford, CA 94305, USA. [4]In Vitro Biosafety Level 3 (BSL-3) Service Center, School of Medicine; Stanford University, Stanford, CA 94305, USA. [5]Sarafan ChEM-H; Stanford University, Stanford, CA 94305, USA. [6]Chan Zuckerberg Biohub – San Francisco, San Francisco, CA 94158, USA. [7]These authors contributed equally: Leonid Andronov, Mengting Han. ✉e-mail: sqi@stanford.edu; wmoerner@stanford.edu

filaments in vesicles[4,5], further investigations are needed to identify specific viral RNAs in the cellular context.

Fluorescence microscopy offers a highly useful and complementary set of capabilities, most importantly the specific labeling of proteins or RNA sequences. However, conventional diffraction-limited (DL) fluorescence microscopy, with its resolution constrained to ~250 nm, is unable to resolve the tiny structures that are hidden in a blurred DL image. Super-resolution (SR) microscopy based on single molecules (PALM[6], (d)STORM[7,8]) or on structured patterns of molecular depletion (STED[9], SIM[10]), however, offers far better optical resolution down to 10 nm and below. A wealth of important cellular patterns and structures has been identified in recent years, such as the banding patterns of axonal proteins in neuronal cells[11] and many others[12–14]. The specificity of SR imaging is useful to apply to the study of viral genomic RNA (vgRNA) and other RNA molecules; moreover, additional nanoscale imaging of critical proteins involved in coronavirus infection of cells provides crucial context for the nearby partners and surroundings of the viral RNA. In a previous proof-of-principle study, we explored the relatively safe human coronavirus 229E (HCoV-229E) from the alphacoronavirus family, which uses the APN receptor and produces only mild cold symptoms[15].

In this work, we apply multicolor confocal microscopy and SR microscopy to explore the localization patterns of viral RNA, related viral proteins, and altered host cell structures for SARS-CoV-2 betacoronavirus during the early and late infection of mammalian cells. The SARS-CoV-2 life cycle starts with viral entry into a host cell, facilitated by binding of viral spike protein to its canonical receptor at the cell surface, the angiotensin-converting enzyme 2 (ACE2)[16], or one of the alternative receptors[17]. The subsequent fusion of the viral and the host cell membranes releases the viral genetic material, positive-sense single-stranded viral genomic RNA (vgRNA), into the cytoplasm, where it is readily translated by host ribosomes. SARS-CoV-2 vgRNA (Fig. 1a) encodes at least 29 proteins, including structural proteins that make up the virions, and non-structural proteins (NSPs) and accessory proteins that exist only within host cells and regulate various processes in the intracellular viral life cycle. All NSPs originate from polyproteins that are translated directly from vgRNA and are self-cleaved by viral proteases. Structural and accessory proteins are translated from shorter viral genome fragments called subgenomic RNAs (sgRNAs) that are transcribed from vgRNA.

Replication and transcription of the viral genome is carried out by the RNA-dependent RNA polymerase complex (RdRp), which is assembled from nsp12 (RdRp catalytic subunit) along with nsp7 and nsp8 (accessory subunits)[18]. RdRp first synthesizes either a full-length negative-sense copy of vgRNA or a subgenomic negative-sense copy of vgRNA, producing double-stranded RNA (dsRNA) that forms between vgRNA and the negative-sense copy. Next, using this negative-sense template, a new vgRNA or an sgRNA is generated by the same polymerase enzyme. Additional NSPs modify newly synthesized viral RNAs to form 5′ cap structures[19] that mimic cellular mRNAs to be translated by host ribosomes. The replication intermediates, such as dsRNA and uncapped RNAs, might be degraded or trigger innate immune response[20] and therefore need to be protected from cellular machinery. SARS-CoV-2 transforms host ER into DMVs[21] that are abundant in the perinuclear region of infected cells[4,5,22] and likely encapsulate dsRNA[3,5] and newly synthesized viral RNA[4,23]. However, the precise intracellular localization of replicating RdRp enzymes and therefore of the replication events is not well established to date[3,23,24].

In this work, we focus particularly on vgRNA, dsRNA and key RdRp subunits nsp12, nsp7 and nsp8. We also co-image a series of molecules, including membrane markers, nucleocapsid protein, spike protein, and the nsp3 protein (reported to be a major component of a molecular pore spanning both membranes of DMVs[25]), all to provide context and support for the view that vgRNA, dsRNA, and RdRp act spatially in replication organelles (ROs) during viral replication. Thus,

we provide key information about where these important players are found in infected cells and how they change with time during early vs late infection. Our results yield a nanoscale optical readout of viral nucleic acid organization and viral proteins down to 20–40 nm during SARS-CoV-2 infection, highlight the structural importance of ROs, and could potentially benefit development of future therapeutic approaches.

## Results

### Labeling and imaging of SARS-CoV-2 virions
To specifically detect SARS-CoV-2 vgRNA, we applied RNA fluorescence in situ hybridization (RNA FISH) with 48 antisense DNA oligonucleotide probes[26] specifically targeting the open reading frame 1a (ORF1a) region which is only present in vgRNA and not in subgenomic RNAs (sgRNAs), ensuring detection of only full-length viral positive-sense vgRNA (Fig. 1a). Each probe was conjugated with a single blinking fluorophore for (d)STORM (direct Stochastic Optical Reconstruction Microscopy)[8]. To test this labeling and imaging approach, we first imaged vgRNA along with SARS-CoV-2 spike protein in purified virions (Supplementary Fig. S1). While the size of SARS-CoV-2 virions is too small to resolve in conventional DL fluorescence microscopy (Supplementary Fig. S1a), in SR the internal concentric organization of the virions can be observed with vgRNA found in their center and spike at the surface (Supplementary Fig. S1b). The labeling efficiency with these probes is around 6 dyes/vgRNA in partially Proteinase K-digested virions, which was higher than in intact virions (1.7 dyes/vgRNA) due to poorer accessibility of their vgRNA (Supplementary Fig. S1c–i).

Next, we imaged SARS-CoV-2 infected Vero E6 cells that were fixed at 24 h post infection (hpi) and then labeled for immunofluorescence imaging (Methods). SR microscopy of spike and nucleocapsid proteins in these cells revealed assembled virions mostly at the cellular periphery, often at cytoplasmic tubular projections of the plasma membrane, indicating active viral production (Supplementary Fig. S1j), similar to previously reported results[5,27]. These studies of virions highlight the improved resolution of SR microscopy and validate the labeling approach, but much more is to be learned by imaging viral RNA and proteins in the cellular interior. We now turn to the main focus of this study, the intracellular replication of viral genomic RNA.

### SARS-CoV-2 genomic RNA clusters in cytoplasm of infected cells
Confocal screening demonstrated three patterns of intracellular vgRNA localization (Supplementary Fig. S2a): scattered puncta in the cytoplasm (Type 1, Fig. 1b), appearance of bright foci in the perinuclear region (Type 2, Supplementary Fig. S2a), and concentration of vgRNA into large dense structures that occupy most of the perinuclear region (Type 3, Fig. 1f). We find that Type 1 cells were most abundant at 6 hpi, and Type 3 cells at 24 hpi, indicating that the vgRNA localization progresses from Type 1 to Type 3 as infection advances in time (Supplementary Fig. S2b). We also find that the cell-integrated vgRNA FISH signal in infected cells increases 2.2× on average from 6 to 24 hpi (Supplementary Fig. S2c), representing active viral replication and accumulation of vgRNA inside the cells.

The higher spatial resolution of SR microscopy revealed that at 6 hpi (Type 1 and Type 2 cells), most vgRNA localizes into clusters with an approximately round shape and a diameter of 100–250 nm that scatter in the cytoplasm (Fig. 1c, d). At 24 hpi (Type 2 and Type 3 cells), the vgRNA localization pattern transformed into a fascinating dense perinuclear network of approximately round structures with a diameter of 300–700 nm (Fig. 1g, h). The lower number of localizations in many cluster centers suggests a possibly hollow structure in these 2D images; future 3D imaging can explore more aspects of the cellular localization. To quantify the transformation of vgRNA clusters in infected cells, we performed a clustering analysis using a Bayesian Information Criterion-optimized Gaussian Mixture Model (BIC-GMM)

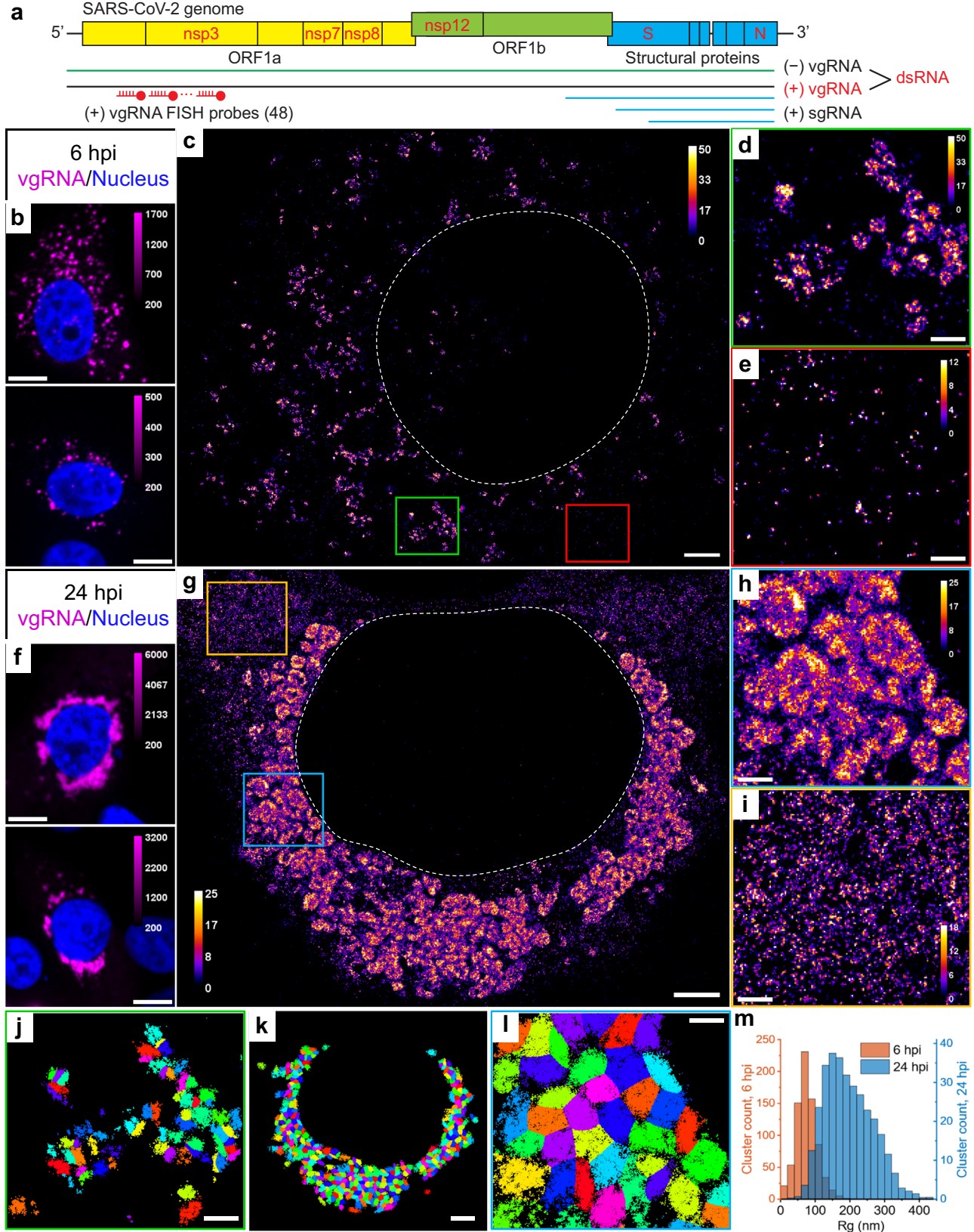

(Fig. 1j–l; see Methods). This analysis showed an increase in the median vgRNA cluster size (Rg, radius of gyration) from 73 nm at 6 hpi to 187 nm at 24 hpi (Fig. 1m), reflecting the drastic change in vgRNA localization pattern.

Besides dense vgRNA clusters, we observe isolated localizations of individual vgRNA molecules scattered in the cytoplasm at both time points, in line with previously reported results[15,26]. These appear as a haze in confocal images (Supplementary Fig. S2a, Type 3) but are resolved as sparse nanoscale puncta (d < 50 nm) in SR (Fig. 1e, i; Supplementary Fig. S3a) which we assume to be single vgRNA copies (even though the puncta are denser at 24 hpi). Using the average number of single-molecule (SM) localizations per vgRNA punctum as a calibration

**Fig. 1 | Clustering of vgRNA in the cytoplasm of infected cells. a** Scheme of SARS-CoV-2 genome with constructs used for its detection in infected cells. 48 antisense DNA oligonucleotide probes were used to target the ORF1a-coding region of vgRNA that is exclusive to the positive-sense vgRNA and does not occur in the sgRNAs. The RNA FISH probes are conjugated with AF647 or CF568. **b** Representative confocal images of vgRNA in infected Vero E6 cells at 6 hpi display scattered diffraction-limited (DL) puncta. **c** Representative SR image of an infected cell at 6 hpi reveals distinct vgRNA clusters in the cytoplasm. **d** Zoomed-in region of the SR image (green frame in **c**) displays an agglomeration of vgRNA clusters. **e** Zoomed-in region of the SR image (red frame in **c**) shows nanoscale puncta of individual vgRNA molecules. **f** Representative confocal images of vgRNA in infected Vero E6 cells at 24 hpi display large DL foci in the perinuclear region of the cytoplasm. **g** Representative SR image of an infected cell at 24 hpi reveals large perinuclear vgRNA clusters. **h** Zoomed-in region of the SR image (blue frame in **g**) displays dense vgRNA clusters. **i** Zoomed-in region of the SR image (yellow frame in **g**) displays nanoscale puncta of vgRNA molecules. **j** BIC-GMM cluster analysis of the region shown in **d**. **k** BIC-GMM cluster analysis of the cell shown in **g**. **l** BIC-GMM cluster analysis of the region shown in **h**. **m** Histogram of the radii of gyration (Rg) of the vgRNA clusters indicate their size increase between 6 hpi (tan) and 24 hpi (blue). Data for the histograms are obtained from 10 cells (6 hpi) and 16 cells (24 hpi) and 2 independent experiments for each time point. Scale bars, 10 μm (**b**, **f**), 2 μm (**c**, **g**, **k**), 500 nm (**d**, **e**, **h**, **i**, **j**, **l**). Dashed lines in **c** and **g** indicate the boundary of the cell nucleus (large dark region). Localizations that belong to the same cluster in **j**, **k**, **l** are depicted with the same color, but colors are reused. Color bars in **c**–**e**, **g**–**i** show the number of single-molecule localizations within each SR pixel (20 × 20 nm$^2$).

for the number of localizations per single vgRNA, we estimated the average number of vgRNA molecules in the vgRNA clusters to be around 26 vgRNA/cluster at 6 hpi, increasing by almost an order of magnitude to 181 vgRNA/cluster at 24 hpi (Supplementary Fig. S3b, c; procedure detailed in Methods).

### dsRNA associates with vgRNA clusters

Next, we proceeded to assess the relation of vgRNA cluster locations to viral replication. For this, we immunofluorescently labeled an intermediate of coronavirus replication and transcription, the hybridized dsRNA objects composed of positive-sense vgRNA and negative-sense copy, and co-imaged dsRNA with vgRNA using two-color confocal and SR microscopy. In confocal microscopy, dsRNA labeling was present in all cells with detectable vgRNA FISH fluorescence, including in early infection, demonstrating the high sensitivity of our dsRNA immunofluorescence detection (Supplementary Fig. S2d). dsRNA and vgRNA appeared mostly colocalized at both time points at low resolution (Fig. 2a, b), suggesting that vgRNA clusters are often found close to the replication centers of SARS-CoV-2. SR microscopy revealed that dsRNA aggregates into clusters of a relatively compact size ($d \approx 100$–200 nm) with distinct patterns of colocalization with vgRNA at 6 or 24 hpi (Fig. 2c, d).

To quantify the spatial relationship between dsRNA and vgRNA, we conducted pair-pair correlation analysis[28]. We calculated a bivariate pair-correlation function $g_{12}(r)$, i.e., the distribution of the pairwise distances between the localizations of the two species[29]. The function is computed only in perinuclear regions and is normalized in a way that $g_{12}(r) = 1$ for two randomly and homogeneously distributed species without interaction, signifying complete spatial randomness (CSR). Closely associated or colocalized species have a prevalence of short pairwise distances resulting in a peak in $g_{12}(r)$ near $r = 0$, while anti-correlated species lack short interparticle distances, which lowers $g_{12}(r)$ at $r = 0$ followed by peaking at $r > 0$.

At early infection stages (6 hpi), dsRNA clusters appear closely associated with or adjacent to vgRNA clusters both visually and by pair-pair correlation analysis (Fig. 2c, e). By contrast, during late infection (24 hpi), dsRNA clusters anticorrelate with vgRNA at short distance scales with an average separation between them around 120 nm as indicated by bivariate pair-correlation functions $g_{12}(r)$ (Fig. 2h). Moreover, at 24 hpi, dsRNA clusters can often be found in the central voids of the large vgRNA structures (Fig. 2d), suggesting their possible concentric localization in the same ROs.

Contrary to vgRNA, the size of dsRNA clusters slightly decreases (Fig. 2f, g) and the total brightness of cellular dsRNA labeling does not significantly change between 6 hpi and 24 hpi (Supplementary Fig. S2e). Interestingly, at 6 hpi but not at 24 hpi, the dsRNA signal per cell positively correlates with that of vgRNA (Supplementary Fig. S2f, g). These findings indicate that the amount of dsRNA increases at early infection but reaches saturation by 24 hpi. This may suggest that after the rapid initial production of a dsRNA pool, further generation of negative-sense copies slows down and the replication shifts to the generation of vgRNA from the pool of available negative-sense templates, which is common in other coronaviruses[30].

### vgRNA clusters denote the replication centers of SARS-CoV-2 genome

To investigate SARS-CoV-2 replication activity within the vgRNA clusters in more detail, we co-imaged them with the RdRp complex, the replicating SARS-CoV-2 RNA-dependent RNA polymerase[18,31], using immunofluorescent labeling of its catalytic subunit nsp12[32]. In confocal images, nsp12 adopts a similar pattern as vgRNA, colocalizing with it at both 6 hpi and 24 hpi (Fig. 3a, b), which suggests ongoing replication at the vgRNA clusters. In SR images, nsp12 localized in small sparse puncta ($d < 50$ nm) that were scattered within and next to the vgRNA clusters at both time points (Fig. 3c, d). Because nanoscale nsp12 puncta are well separated from each other, and oligomerization is not expected[18,31,33], each nanoscale punctum is likely to represent a single replicating enzyme. On average, we detected 2.5 nsp12 puncta per vgRNA cluster at 6 hpi and 7.6 at 24 hpi (Fig. 3h).

From comparison of DL and SR images, one may infer fundamentally different (large-scale) nsp12 structures at 6 hpi and 24 hpi in confocal microscopy (Fig. 3a, b). In DL microscopy, ROs do look like individual diffraction-limited dots at 6 hpi when they are sparse (Fig. 3a), i.e., the average distance between them is larger than the diffraction limit (even though the individual RdRp complexes inside ROs are still not resolved). The same organelles when they are dense at 24 hpi resemble large irregular blobs because the distance between the individual organelles becomes smaller than the diffraction limit (Fig. 3b). This filling in with optically overlapping ROs creates a misleading perception of distinct structures in confocal microscopy. However, SR microscopy, which sees spatial details on the scale of 20–40 nm, resolves both types of structures much better. The nsp12 puncta are small in both cases because they arise from individual RdRp enzymes, yet the vgRNA clusters are smaller at 6 hpi and larger at 24 hpi, which is a better representation of the size of these assemblies.

Therefore, in contrast to vgRNA but similar to dsRNA, the total cellular amount of nsp12 does not significantly increase (Supplementary Fig. S2h) and its nanoscale localization pattern stays the same as infection progresses from 6 to 24 hpi (Fig. 3c, d). This suggests that the growth of vgRNA clusters arises from a relatively constant small number of replication components between 6 and 24 hpi highlighted by the constant amount of dsRNA and RdRp. Bivariate cross-correlation functions calculated between nsp12 and vgRNA localizations peaked at 0 nm indicating association of these two targets at both 6 and 24 hpi (Fig. 3i). Since vgRNA clusters colocalize with the catalytic subunit of RdRp, we suggest that vgRNA clusters combined with the nearby RdRp enzymes and dsRNA highlight ROs that act as centers for replication and transcription of SARS-CoV-2.

To verify that nsp12 labeling is a good reporter of assembled replication complexes, we have also imaged two accessory subunits of RdRp, nsp7 and nsp8. We find close association of these subunits with vgRNA as shown in Fig. 3e, f, and in the pair-correlation functions of

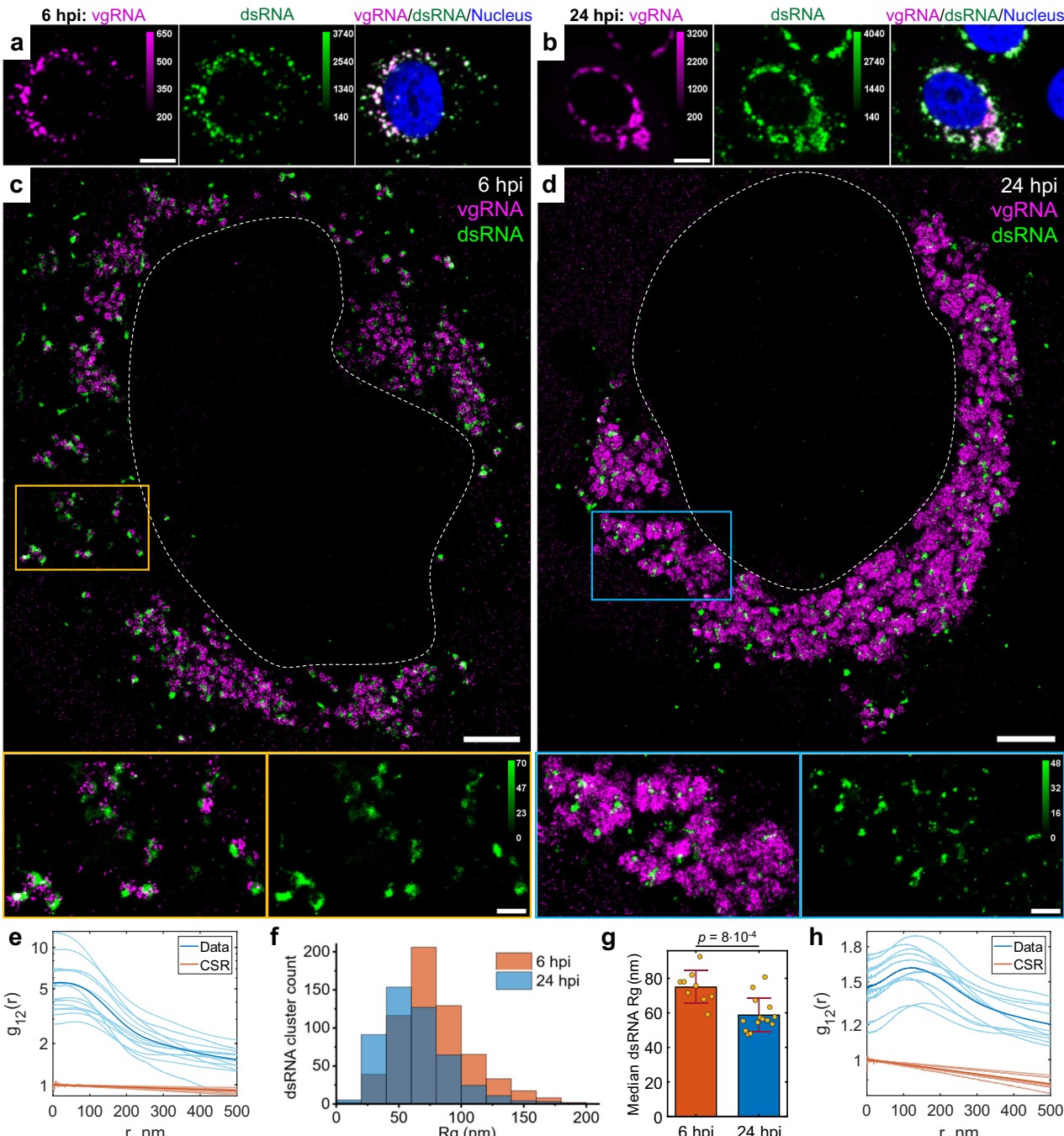

**Fig. 2 | Association of dsRNA with vgRNA clusters.** Representative confocal images of SARS-CoV-2 infected cells display DL colocalization between dsRNA (green) and vgRNA (magenta) at both 6 hpi (**a**) and 24 hpi (**b**). Representative SR images of SARS-CoV-2 infected cells indicate association between dsRNA and vgRNA at 6 hpi (**c**) and short-range anti-correlation often with concentric localization at 24 hpi (**d**). Bottom panels, zoomed-in images of corresponding colored boxes. **e** Bivariate pair-correlation functions $g_{12}(r)$ calculated between the localizations of dsRNA and vgRNA indicate their close association at 6 hpi. **f** Histogram of Rg of dsRNA clusters as determined by the BIC-GMM cluster analysis. **g** Median Rg

of dsRNA clusters per cell significantly decreases between 6 hpi and 24 hpi. Data points denote median Rg values of dsRNA clusters in individual analyzed cells; error bars represent mean ± standard deviation of these values. Data for the histograms (**f**, **g**) are obtained from 9 cells (6 hpi) and 14 cells (24 hpi). **h** Bivariate pair-correlation functions $g_{12}(r)$ reveal nanoscale anti-correlation between dsRNA and vgRNA at 24 hpi. CSR, complete spatial randomness. Thin lines correspond to $g_{12}(r)$ of individual cells and bold lines represent the mean values of $g_{12}(r)$ from all analyzed cells. Scale bars, 10 μm (**a**, **b**), 2 μm (**c**, **d**), 500 nm (**c**, **d**, bottom panels). Dashed lines in **c** and **d** indicate the boundary of the cell nucleus.

Fig. 3i (see also Supplementary Figs. S4 and S5). Nsp12 and nsp8 colocalized with each other on the nanoscale (Supplementary Fig. S6), indicating their interaction within ROs, as expected for subunits of assembled RdRp.

Finally, to confirm that the vgRNA clusters we observe contain newly replicated viral RNA, we provided brominated uridine (BrU) to the infected cells in the form of 5-bromouridine 5′-triphosphate

(BrUTP) for 1 h before fixation while endogenous transcription was inhibited by actinomycin D[23,34,35]. Immunofluorescent labeling of BrU then highlights newly replicated RNA. Confocal and SR imaging localizes RNA-containing BrU to the perinuclear clusters of vgRNA (Fig. 3g, Supplementary Fig. S7) and close to nsp12 (Supplementary Fig. S8), further proving that these structures are the sites of active replication and transcription of viral RNA.

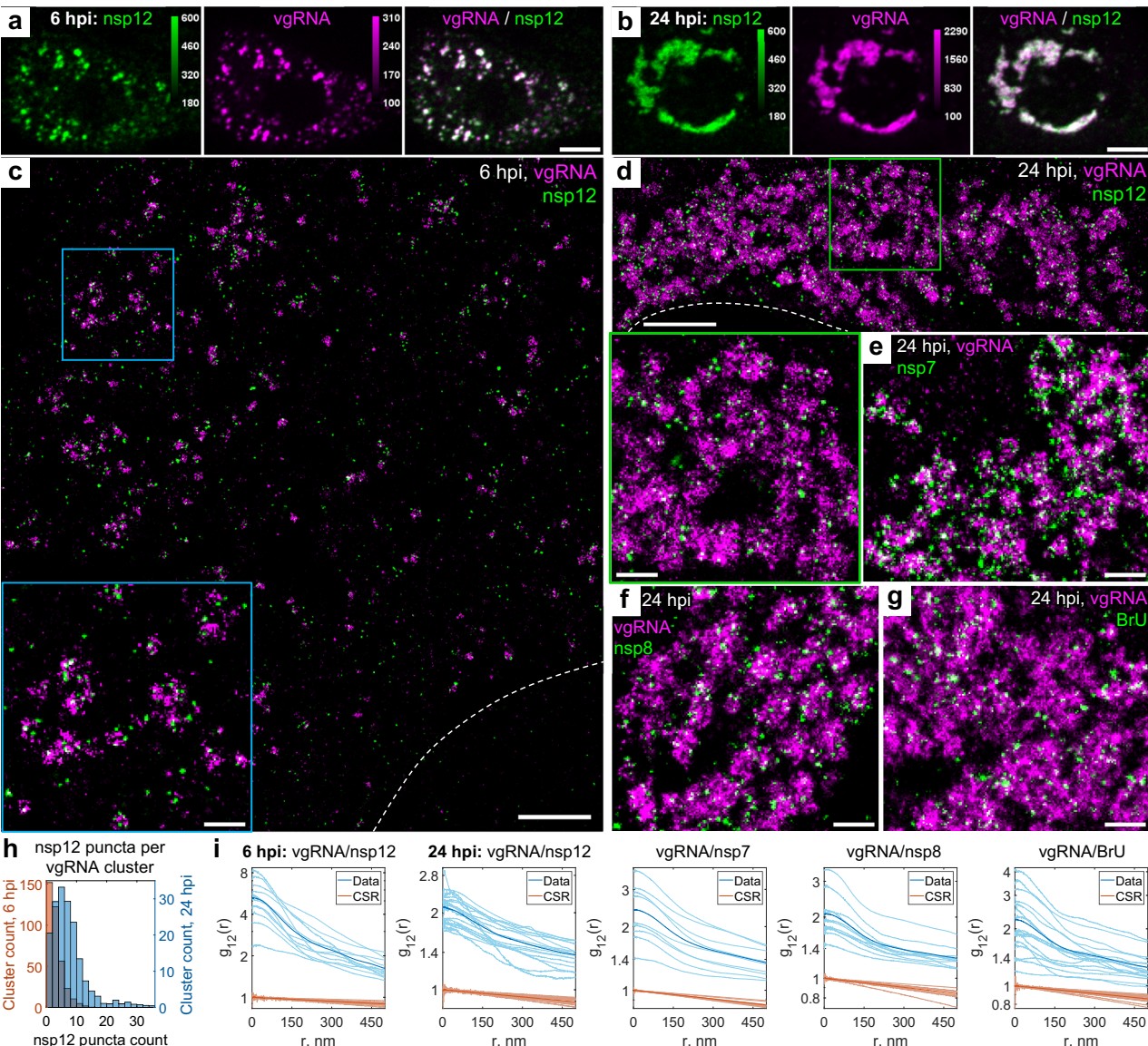

**Fig. 3 | Association of SARS-CoV-2 replication enzyme with vgRNA clusters.**
Representative confocal images of SARS-CoV-2 infected cells display DL colocalization between nsp12, the catalytic subunit of RdRp (green) and vgRNA (magenta) at both 6 hpi (**a**) and 24 hpi (**b**). Representative SR images of SARS-CoV−2 infected cells indicate nanoscale association between nsp12 and vgRNA at both 6 hpi (**c**) and 24 hpi (**d**). Insets show magnified images of corresponding regions in the colored boxes. Representative SR images of vgRNA with nsp7 (**e**) or nsp8 (**f**) in the perinuclear regions of SARS-CoV-2 infected cells indicate association of nsp7 and nsp8 with vgRNA clusters. **g** Representative SR image of vgRNA with newly synthesized viral RNAs labeled by BrU in a SARS-CoV-2 infected cell indicates localization of newly synthesized viral RNAs within the perinuclear clusters of vgRNA. **h** Number of nanoscale puncta of nsp12 per vgRNA cluster. Data for the histograms are obtained from 5 cells (6 hpi) and 9 cells (24 hpi). **i** Bivariate pair-correlation functions for vgRNA and nsp12, nsp7, nsp8 and newly transcribed viral RNA labeled with BrU peak at $r = 0$ nm indicating association between these target pairs. Scale bars, 10 μm (**a**, **b**), 2 μm (**c**, **d**), 500 nm (**e**–**g** and insets in **c** and **d**). Dashed lines in **c** and **d** indicate the edge of the cell nucleus.

## vgRNA clusters are enclosed in ER-derived membranous organelles

Coronaviruses are known to transform the host ER into replication-permissive structures, such as convoluted membranes and DMVs[3,22,36]. To investigate the relation of vgRNA clusters with cellular ER, we immunofluorescently labeled Sec61β, an ER membrane protein[37], in Vero E6 cells stably expressing Sec61β-GFP[15]. Confocal images of these cells show the appearance of Sec61β spots that colocalize with vgRNA against the mostly unaltered ER background at 6 hpi (Fig. 4a). At 24 hpi, however, substantial amounts of Sec61β accumulate close to the perinuclear vgRNA clusters, while the ER tubules outside these regions become poorly visible (Fig. 4a), consistent with the virus-induced rearrangement of the ER and the inhibition of host gene expression by SARS-CoV-2[38].

In SR, we observe encapsulation of the vgRNA clusters by ring-like structures of the altered ER at 6 hpi (Fig. 4b, Supplementary Fig. S9). As infection progresses, the ER-derived ring- or sphere-like structures grow to accommodate larger vgRNA clusters at 24 hpi (Fig. 4b, Supplementary Fig. S10). Pair-correlation functions peak at the distance of the typical radius of vgRNA clusters indicating nanoscale anti-correlation compatible with the ER-derived encapsulation of vgRNA (Fig. 4c). dsRNA (Fig. 4b, Supplementary Fig. S11) and nsp12 (Fig. 4b) are also found to be encapsulated by the same remodeled ER

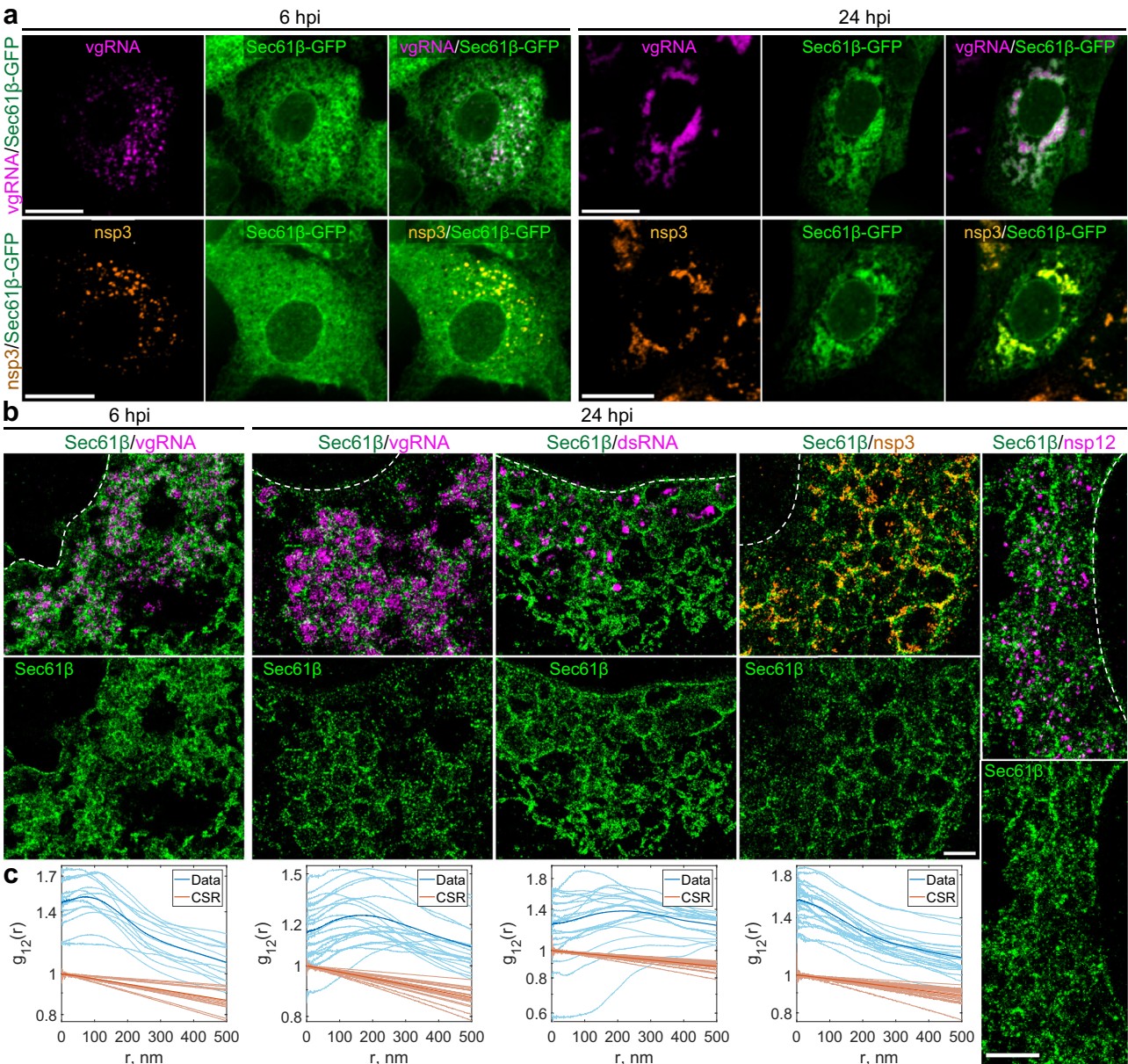

**Fig. 4 | vgRNA clusters are encapsulated in membranes of remodeled ER.**
**a** Representative confocal images of SARS-CoV-2 infected cells indicate an appearance of dense perinuclear foci of Sec61β ER labeling (green) at 24 hpi that colocalizes with vgRNA and nsp3. **b** SR images reveal concentric organization of Sec61β around vgRNA, dsRNA and nsp12, and colocalization of Sec61β with nsp3. **c** Bivariate pair-correlation functions indicate anti-correlation of Sec61β with vgRNA and dsRNA and association of Sec61β with nsp3. Scale bars, 20 μm (**a**) and 1 μm (**b**). Dashed lines in **b** indicate the boundary of the cell nucleus.

membranes suggesting that vgRNA, dsRNA and RdRp are all located within the same ER-derived ROs.

To further confirm that these clusters are surrounded by membranes, we used a (d)STORM-compatible general membrane marker CellMask Deep Red[39]. This dye broadly stains cellular membranes, including the nuclear envelope, mitochondrial membranes, and SARS-CoV-2 virions at the plasma membrane (Supplementary Fig. S12). The nanoscale image contrast with CellMask Deep Red is poorer than specific protein labeling of the Sec61β ER label due to background from membranes of different cellular organelles. Nevertheless, in the perinuclear region of infected cells, we observed the appearance of a complex membranous network that anti-correlates with vgRNA and dsRNA, with visible encapsulation of vgRNA and dsRNA clusters (Supplementary Figs. S12 and S13). Taken together, these findings indicate that each vgRNA-dsRNA-RdRp cluster is located inside a

membrane-bound RO that originates from altered host ER transformed by SARS-CoV-2.

## Nsp3, spike and nucleocapsid proteins localize at the surface of SARS-CoV-2 replication organelles

Because the nsp3 protein of betacoronaviruses is essential for the DMV formation[21,40], and nsp3 is a constituent of a DMV molecular pore[25], we proceeded to localize this non-structural protein to relate the ROs to the SARS-CoV-2-induced DMVs. At DL resolution, nsp3 labeling adopts a pattern that colocalizes with vgRNA at both 6 and 24 hpi, similar to dsRNA and nsp12 (Fig. 5a, f). SR imaging of these cells, however, revealed striking nanoscale positioning of nsp3. At 6 hpi, sparse nsp3 can be found surrounding isolated vgRNA clusters (Fig. 5b, c), while larger nsp3 aggregates are situated amidst bunched vgRNA clusters (Fig. 5d). At 24 hpi, nsp3 localizes at the borders of the large vgRNA

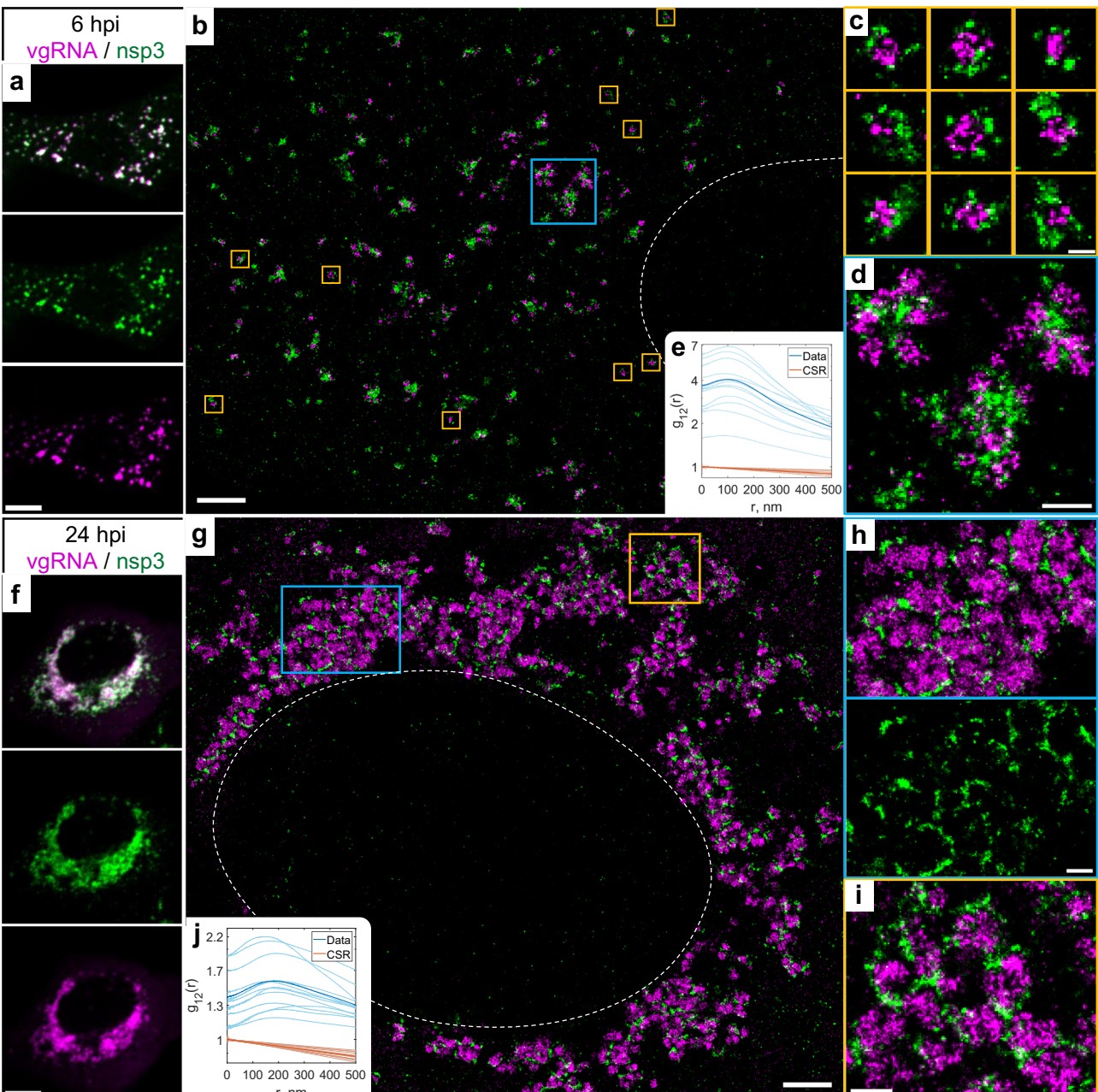

**Fig. 5 | Nsp3 localizes at the surface of vgRNA clusters. a** Representative confocal images of a SARS-CoV-2 infected cell display DL colocalization between punctate vgRNA (magenta) and nsp3 (green) labeling at 6 hpi. **b** Representative SR image of a SARS-CoV-2 infected cell at 6 hpi. **c** Zoomed-in images of selected vgRNA particles (yellow boxes in **b**) indicate the localization of nsp3 at the surface of the vgRNA clusters. **d** Magnified region with aggregates of vgRNA clusters (blue box in **b**) displays dense nsp3 localization in the core of these aggregates. **e** Bivariate pair-correlation functions calculated between the SM localizations of vgRNA and nsp3 indicate nanoscale anti-correlation of these targets at 6 hpi. **f** Confocal images show that vgRNA and nsp3 occupy approximately the same regions in a SARS-CoV-2 infected cell at 24 hpi. **g** Representative SR image of a SARS-CoV-2 infected cell at 24 hpi. **h, i** Magnified regions of the SR image (colored boxes in **g**) reveal that nsp3 localizes in interstitial regions or encapsulates vgRNA clusters. **j** Bivariate pair-correlation functions indicate nanoscale anti-correlation between vgRNA and nsp3 at 24hpi. Scale bars, 10 μm (**a, f**), 2 μm (**b, g**), 500 nm (**d, h, i**), 200 nm (**c**). Dashed lines in **b** and **g** indicate the boundary of the cell nucleus.

clusters, encircling them in incomplete rings and forming a partial perinuclear network (Fig. 5g–i). Similar nsp3 arrangements can be observed in relation to dsRNA (Supplementary Fig. S14).

The anti-correlation of vgRNA with nsp3 and dsRNA with nsp3 (Fig. 5, Supplementary Fig. S14) closely resemble the pattern observed with vgRNA and dsRNA with Sec61β (Fig. 4, Supplementary Fig. S11), suggesting that nsp3 may also be localized at the ER-derived membranous surface of the ROs. To further confirm this hypothesis, we co-imaged nsp3 with Sec61β and CellMask (Fig. 4, Supplementary Figs. S13, S15). The SR images and the pair-correlation analysis

indicated colocalization between nsp3 and both membrane markers at both time points (Fig. 4b, c, Supplementary Figs. S13, S15), confirming that nsp3 localizes on the membranes encircling the SARS-CoV-2 ROs.

Besides these characteristic localization patterns of nsp3, we observed a few cells with two different phenotypes at 24 hpi, one with an ER-like network that occupies large regions in the cytoplasm (Supplementary Fig. S16a), and another one with nsp3 densely diffused throughout the whole cytoplasm (Supplementary Fig. S16b). The ER-like network may represent nsp3 proteins being heavily translated on ER membranes, while nsp3 proteins found outside the perinuclear

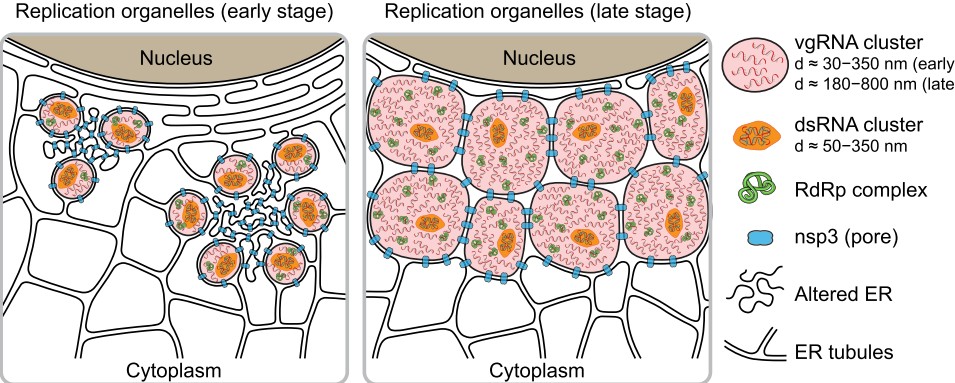

**Fig. 6 | Proposed model for SARS-CoV-2 replication organelles.** Replication organelles (ROs) comprise various RNA species (vgRNA, dsRNA) and proteins (RdRp, nsp3) organized by remodeled cellular ER membranes. Our data support a model which specifies the spatial organization of ROs at early (left panel) and late (right panel) stages of infection. Specific RNA and protein molecules are shown to be distinctly organized within the ROs. The replication and transcription of SARS-CoV-2 RNA occur within the double-membrane vesicles (DMVs) and vesicle packets (VPs) in proximity to RdRp.

region are less likely to be associated with the SARS-CoV-2 replication process and might represent other nsp3 functions, such as a papain-like proteolytic function[41] or post-translational modification of host proteins[42], which can become objects of future SR studies.

The localization of nsp3 at the surface of isolated vgRNA-dsRNA clusters at 6 hpi is consistent with the localization of molecular pores on the DMV membrane observed by cryo-EM[25]. At late infection times, DMVs have been observed to merge into vesicle packets (VPs)[5] that are also likely to contain pores, however molecular pores in the VP membranes have not yet been studied in detail to our knowledge. Nevertheless, previous studies report that in late infection the perinuclear region becomes filled with DMVs and VPs[22] that strongly resemble the ROs reported here. The size of vgRNA clusters at 6 hpi and at 24 hpi from our data is similar to the previously reported size of DMVs and VPs, correspondingly[5].

To search for a possible role of perinuclear vgRNA clusters in virion assembly, we co-imaged vgRNA with two SARS-CoV-2 structural proteins, spike and nucleocapsid (Supplementary Figs. S17, S18). Spike labeling forms typical ~150 nm hollow particles at the cell periphery, and we detect weak vgRNA signal in the center of some of these particles (Supplementary Fig. S17b), consistent with the structure of SARS-CoV-2 virions that contain a single vgRNA molecule. Inside the host cells, spike localizes at the nuclear envelope and in some cytoplasmic organelles; however, it is mostly excluded from the perinuclear vgRNA clusters (Supplementary Fig. S17a, c). Nucleocapsid protein demonstrates rather diffuse localization throughout the cytoplasm, in accordance with its function in the formation of SARS-CoV-2 ribonucleocapsid complexes[43], but is also excluded from the RO interior (Supplementary Fig. S18a). Nevertheless, in the perinuclear region we detect sparse localizations of both spike and nucleocapsid proteins next to the vgRNA clusters and between them, likely at the DMV membranes, as highlighted by anti-correlation of these proteins with vgRNA at $r < 200$ nm (Supplementary Figs. S17c, S18b), similar to the nsp3/vgRNA and Sec61β/vgRNA pairs. The localization of nucleocapsid protein at the RO membranes has already been reported[44], and spike protein has a transmembrane domain[45] and tends to localize not only to virion membranes, but also to intracellular membranes, such as the nuclear envelope (Supplementary Fig. S17a); therefore, small amounts of spike can also be present at RO membranes. Our SR data suggests that while the vgRNA clusters are not directly involved in SARS-CoV-2 virion assembly, it is possible that early stages of virion assembly start at the RO membrane, once vgRNA molecules leave the ROs.

Taken together, our results provide evidence that vgRNA accumulates in DMVs at 6 hpi and in VPs at 24 hpi. dsRNA clusters occur within the same vesicles but occupy distinct parts of them compared to vgRNA. Our data suggests a model (Fig. 6) where SARS-CoV-2 RNA is replicated and transcribed within these DMVs and VPs as highlighted by the proximal localizations of RdRp.

## Discussion

Previous biochemical and EM studies allowed researchers to build models of the intracellular life cycle of SARS-CoV-2[24,46,47]; however, precise localization of specific viral proteins and RNA molecules is challenging due to lack of specific contrast in EM and low resolution in DL fluorescence microscopy. SR fluorescence microscopy is well suited for coronavirus studies in cells as it provides both specific contrast and high resolution (~20 nm and below depending upon photons collected[48]). However, to date few studies have employed this method for coronavirus biology[15], with even less focus on SARS-CoV-2[36,44,49], and none of them addressed the SARS-CoV-2 replication process in detail. Here we apply SR fluorescence microscopy to precisely localize the key players of SARS-CoV-2 replication at different time points in infected cells. Building upon a previously developed method for simultaneous labeling of coronavirus vgRNA with dsRNA and protein immunofluorescence[15], and using improved fixation and multi-color SR imaging protocols (see Methods), we obtain and quantify the appearance and molecular compositions of ROs of SARS-CoV-2 in cells at different stages of infection.

In this study, our results taken together depict a compelling and novel picture of ROs containing various molecules including vgRNA, dsRNA, RdRp, nsp3, and ER membrane (Fig. 6). In this model, we compare the organization of ROs at early and late stages of infection and show how specific RNA and protein molecules are spatially organized in ROs. Compared to the simpler and less pathogenic HCoV-229E case, SARS-CoV-2 appears to generate more complex clusters of vgRNA, and with the imaging of viral proteins involved in vgRNA replication and in DMV formation, the structural importance of ROs is now clear.

The detailed intracellular localization of the central SARS-CoV-2 component, vgRNA, has remained vague in the literature. Our RNA FISH method[15] targets specific sequences in vgRNA (Fig. 1a) and detects single vgRNA molecules (Fig. 1e, i; Supplementary Figs. S1, S3a, S17b), allowing counting of the number of vgRNA molecules within specific regions (Supplementary Fig. S3b, c). We find for the first time that most cellular vgRNA localizes into dense clusters of an approximately round shape that grow and migrate to the perinuclear region as infection time increases. We show that these clusters appear confined in membranous vesicles derived from ER as emphasized by the localization of Sec61β and CellMask at their surface (Fig. 4b, Supplementary Figs. S9–S13). From comparison with earlier EM images[5,22,25] and from

nsp3 localization at their surface[25] (Fig. 5), we can conclude that these vesicles are most likely DMVs at an early-mid infection time that grow and merge into VPs as infection progresses.

Previously, metabolic radioactive labeling was used to localize newly synthesized RNA in SARS-CoV-1 and MERS-CoV-infected cells to DMVs[4]. However, metabolic labeling only localizes a fraction of vgRNA molecules with little sequence specificity and with a background of viral sgRNA. Here, we specifically label vgRNA of SARS-CoV-2 for SR microscopy and show that it also localizes in patterns that suggest confinement in DMVs, consistent with the earlier findings on SARS-CoV-1 and MERS-CoV[4]. Our metabolic labeling of infected cells with BrUTP also localized newly synthesized viral RNAs to the perinuclear vgRNA clusters (Fig. 3g, Supplementary Fig. S7), which agrees with earlier results[4,23,34] and solidifies our conclusions on the spatial localization of vgRNA and viral replication machinery inside DMVs.

Previous studies also suggested the presence of dsRNA in DMVs of SARS-CoV-1[3] and SARS-CoV-2[5]. EM images of DMVs often display a complex filamentous network in their interior that was attributed to viral RNA molecules[5]. However, the exact type of these RNAs was not determined due to the absence of specific labeling. As one might expect, single-stranded vgRNA can form a secondary structure that includes many short dsRNA fragments e.g., in stem loops[50,51]. This makes it difficult to distinguish between viral dsRNA and vgRNA by measuring the diameter of the filaments, taking into account that the detection probability of ssRNA might be lower due to a decreased EM contrast for ssRNA than for dsRNA. Reported abundant branching of filaments in DMVs[5], however, is typical for ssRNA secondary structures[52]. Indeed, these references present some evidence about the presence of both dsRNA and vgRNA in DMVs; however, to our knowledge, there was no simultaneous observation of both vgRNA and dsRNA within the same DMVs.

Here we use the J2 anti-dsRNA antibody that recognizes only long dsRNA fragments (≥40 bp) with no detection of the ssRNA secondary structures[53,54]. The J2 antibody has been reported to underestimate dsRNA localization[26]; however, using optimized antibody concentrations (Supplementary Figs. S19, S20) and optimized staining protocols as detailed in Methods, we achieved excellent sensitivity to dsRNA with signal present in all infected cells, even in early infection with very low vgRNA levels (Supplementary Fig. S2d). Our two-color SR imaging revealed for the first time that most dsRNA and vgRNA are located within the same DMVs and VPs, occupying distinct regions of these vesicles, and adopting an anti-correlation pattern at short distances ($r < 100$ nm) at 24 hpi (Fig. 2). Another novel observation is the relatively constant amount of dsRNA and a slight decrease in dsRNA cluster size between 6 and 24 hpi despite the huge change in the vgRNA landscape (Fig. 2, Supplementary Fig. S2c, e).

It has been proposed that the RdRp complex of SARS-CoV-1 is located at convoluted membranes and inside DMVs based on immunogold labeling of nsp8[3]. However, nsp8 has intracellular functions other than as an RdRp accessory subunit[55,56] that might be exercised at the convoluted membranes. Here we label the catalytic RdRp subunit, nsp12[18], and find that it mostly localizes to the vgRNA clusters at both 6 and 24 hpi (Fig. 3a–d, i), suggesting that SARS-CoV-2 replication and transcription occur preferentially in the vgRNA-filled ROs, where dsRNA resides as well. Additional experiments revealed that two other RdRp subunits, nsp7 and nsp8, as well as newly synthesized viral RNA also localize to the vgRNA clusters (Fig. 3e–g, i; Supplementary Figs. S4–S8), further proving the role of these clusters as replication organelles.

Nsp3 of betacoronaviruses (SARS-CoV-1, MERS-CoV and MHV) was previously localized to the convoluted membranes and to the DMV membranes using immuno-EM[3,4,57,58] and cryo-ET[25]; however, these studies were limited to early-mid infection at 8-12 hpi. In our study, we report two localization patterns of nsp3 of SARS-CoV-2 at 6 hpi: 1) sparse nsp3 at the surface of isolated vgRNA-dsRNA clusters (Fig. 5c, Supplementary Fig. S14a); and 2) dense nsp3 within the accumulations of vgRNA-dsRNA clusters (Fig. 5d, Supplementary Fig. S14a). While the first pattern most likely corresponds to the RO/DMV membranes considering the role of nsp3 as a DMV pore[25], the second one resembles a pattern found in other coronaviruses that was attributed to the convoluted membranes[3,4,59]. Convoluted membranes are typically found within dense groups of DMVs in early-mid infection[3,4] and localization of nsp3 on them might represent early steps of viral transformation of ER into DMVs. We found this nsp3 pattern anti-correlated with vgRNA (Fig. 5d, e) and with dsRNA (Supplementary Fig. S14a, b), suggesting little to no vgRNA or dsRNA at the convoluted membranes, in line with previous studies on other coronaviruses[4].

At 24 hpi, we did not observe these early infection patterns of nsp3 localization. Instead, we show for the first time that at 24 hpi, nsp3 densely localizes at the membranes that separate large vgRNA clusters and grows into a considerable perinuclear network that contains the ROs (Fig. 5g–i, Supplementary Fig. S14c). Since the molecular pores of VPs have not yet been investigated in detail, we can speculate that this late infection nsp3 pattern corresponds to the pores of VPs that should also be much denser than those of isolated DMVs, considering the increased density of nsp3 labeling. Additional rare phenotypes of nsp3 localization that we also report for the first time (Supplementary Fig. S16) illustrate the variability of SARS-CoV-2 infection course and should lead to further research on the other intracellular functions of this viral protein.

Taken together, we investigated several key factors of SARS-CoV-2 replication: vgRNA, dsRNA, RdRp and nsp3 inside infected cells with SR microscopy for the first time. We discovered and characterized the nanoscale structure of perinuclear clusters of vgRNA and demonstrated by RdRp labeling that they associate with SARS-CoV-2 ROs. We found that the ROs also contain dsRNA and are encapsulated in ER-derived membranes. Using SR data on nsp3, we conclude that these virus-induced organelles correspond to DMVs.

This study expands the knowledge of the biology of coronaviruses and opens new possibilities for therapeutics against SARS-CoV-2, considering that clusters of vgRNA have also been reported in SARS-CoV-2 infected interstitial macrophages of human lungs[17], suggesting their importance in COVID-19. Careful examination of the organization of ROs may provide new avenues to target the organelles to disrupt SARS-CoV-2 replication and transcription. Examining localization patterns for different viral variants or in different host cells will be useful to broaden understanding of the viral infection. It will also be important to examine how the structures reported in this study change upon the addition of drug treatments. Our imaging approach may also offer insights into long COVID by investigating cells that are infected by SARS-CoV-2 that may still contain RO-like structures after symptoms disappear.

## Methods

### Antibodies

Primary antibodies and the optimal dilutions and concentrations used are as follows: goat polyclonal anti-spike S2 (Novus Biologicals, AF10774-SP, 1:20, 10 μg/mL), mouse monoclonal anti-dsRNA (J2 clone, SCICONS, 10010200, lot J2-2017, 1:200, 5 μg/mL), rabbit polyclonal anti-nsp12 (Sigma-Aldrich, SAB3501287-100UG, lot 926721011, 1:500, 2 μg/mL), mouse monoclonal anti-nucleocapsid (Thermo Fisher, MA5-29981, lot XF3619701, 1:500, 2 μg/mL), rabbit polyclonal anti-nsp3 (Thermo Fisher, PA5-116947, lot YA3808034B, 1:134, 5 μg/mL), sheep polyclonal anti-GFP (Bio-Rad, 4745-1051, 1:1000, 5 μg/mL), rabbit polyclonal anti-GFP (Novus Biologicals, NB600-308SS, lot 48142, 1:163, 5 μg/mL), rabbit monoclonal anti-nsp7 (GeneTex, GTX636719, lot 44620, 1:200, 2 μg/mL), mouse monoclonal anti-nsp8 (GeneTex, GTX632696, lot 42345, 1:134, 5 μg/mL), mouse monoclonal anti-BrdU (MoBU-1 clone, Thermo Fisher, B35128, lot 2712999, 1:50, 2 μg/mL).

Secondary antibodies and the optimal dilutions and concentrations used are as follows: AF647-conjugated donkey anti-mouse IgG (Thermo Fisher, A-31571, lot 2555690, 1:500, 4 µg/mL), AF647-conjugated donkey anti-rabbit IgG (Thermo Fisher, A-31573, lot 2359136, 1:500, 4 µg/mL), AF647-conjugated donkey anti-sheep IgG (Thermo Fisher, A-21448, lot 2454134, 1:500, 4 µg/mL), CF568-conjugated donkey anti-goat IgG (Sigma-Aldrich, SAB4600074-50UL, lot 18C0723, 1:500, 4 µg/mL), CF568-conjugated donkey anti-rabbit IgG (Sigma-Aldrich, SAB4600076-50UL, lot 21C1025, 1:500, 4 µg/mL), CF568-conjugated donkey anti-mouse IgG (Sigma-Aldrich, SAB4600075-50UL, lot 17C1116, 1:500, 4 µg/mL), CF568-conjugated donkey anti-sheep IgG (Sigma-Aldrich, SAB4600078-50UL, lot 11C1031, 1:500, 4 µg/mL), CF583R-conjugated donkey anti-mouse IgG (Biotium, Custom CF Dye, lot 23C1122, 1:250, 4 µg/mL), CF583R-conjugated donkey anti-rabbit IgG (Biotium, Custom CF Dye, lot 23C0811, 1:250, 4 µg/mL). To confirm that the fluorophore attached to the secondary antibody does not produce artifacts, in a number of cases we switched the labels by switching the secondary antibodies, and found no difference in the SR structures observed.

## Culture of cell lines

The Vero E6 cells (African green monkey kidney epithelial cells, ATCC, CRL-1586), HEK293T cells (human embryonic kidney epithelial cells, ATCC, CRL-3216), and Vero E6-TMPRSS2 cells (BPS Bioscience, 78081) were cultured in Dulbecco's modified Eagle medium (DMEM) with GlutaMAX, 25 mM D-Glucose, and 1 mM sodium pyruvate (Gibco, 10569010) in 10% FBS (Sigma-Aldrich, F0926) at 37 °C and 5% $CO_2$ in a humidified incubator. Cell lines were not authenticated after purchase prior to use. For Vero E6-TMPRSS2, Geneticin (G418) was added at a final concentration of 1 mg/ml.

## Lentivirus production for ER labeling with Sec61β

To produce lentivirus, HEK293T cells were cultured in 10-cm dishes and transiently transfected with 9 µg lentiviral plasmid pLV-ER-GFP (Addgene, 80069, a gift from Pantelis Tsoulfas), 8 µg pCMV-dR8.91, and 1 µg PMD2.G packaging plasmids using 25 µL TransIT-LT1 Transfection Reagent (Mirus, MIR 2306). After 72 h of transfection, supernatant was filtered through 0.45 µm filters, concentrated using Lentivirus Precipitation Solution (ALSTEM, VC100) at 4 °C overnight, and centrifuged at 1500 × g for 30 min at 4 °C to collect virus pellets. The virus pellets were resuspended in cold DMEM for storage at −80 °C for transduction of cells.

## Generation of stable cell line

To generate a Vero E6 cell line stably expressing Sec61β-GFP, 2·10⁵ Vero E6 cells were seeded in one well of a 6-well plate and infected with one quarter of concentrated lentivirus expressing pLV-ER-GFP produced from one 10-cm dish of HEK293T cells while seeding. After two days incubation, monoclonal cells expressing GFP were sorted out using a SONY SH800S sorter. These transduced cells were only used for ER imaging; all other experiments used wild type (WT) cells.

## SARS-CoV-2 viral stocks preparation

The SARS-CoV-2 WA 1, isolate USA-WA1/2020 (NR-52281, BEI Resources) was passaged 3 times in Vero E6-TMPRSS2 cells as previously described[60,61]. Briefly, a Vero E6-TMPRSS2 monolayer was infected with virus obtained from BEI; post 72 h of infection (hpi), P1 virus-containing tissue culture supernatants were collected and stored at −80 °C. Following titration, P1 virus stock was used to generate a P2 stock by infecting Vero E6 TMPRSS2 monolayers with multiplicity of infection (MOI) of 0.0001 for 72 h. P2 virus was passaged again in Vero E6-TMPRSS2 cells to obtain P3 stock. Viral titers were determined by standard plaque assay on Vero E6 cells.

## Infection of cells by SARS-CoV-2

Vero E6 cells previously cultured in 8-well µ-slides (ibidi, 80827-90) were infected in the BSL-3 facility with SARS-CoV-2 WA 1 (USA212 WA1/2020) in triplicates (MOI = 0.5 SARS-CoV-2 WA1 (P3)) at an MOI of 2 for 6 hpi and MOI of 0.2 for 24 hpi. After 6 and 24 h of incubation, cells were washed with PBS and fixed by 4% PFA (Electron Microscopy Sciences #15710) and 0.1% glutaraldehyde (Electron Microscopy Sciences #16350) in PBS for 1 h and removed from BSL-3 for further processing. All work involving viral stock preparation and infection using WT SARS-CoV-2 was conducted at the high containment BSL-3 facility of Stanford University according to CDC and institutional guidelines. All the experiments were performed using a P3 SARS-CoV-2 USA-WA1/2020, containing 100% WT population with no deletion in the spike multi-basic cleavage site.

## Synthesis of the RNA FISH probes

vgRNA FISH probes targeting the ORF1a region of SARS-CoV-2[26] were ordered with 5AmMC6 modifications from Integrated DNA Technologies, Inc. in plate format of 25 nmol scale with standard desalting. The sequences of the 48 probes are as follows[26]: tagatcggcgccgtaactat; gttatcgacatagcgagtgt; agcatccgaacgtttgatga; ccagttgttcggacaaagtg; caaagccacgtacgagcacg; tttcagaacgttccgtgtac; taagctcacgcatgagttca; ggtgacgcaactggatagac; aggttgttctaatggttgta; tgagttggacgtgtgttttc; acaacctatgttagcgctag; tcctttattaccgttcttac; ttatagcggccttctgtaaa; agaagaaccttgcggtaagc; gagcaacataagcccgttaa; ctttctgtacaatccctttg; ctcgtcgcctaagtcaaatg; ttaccagcacgtgctagaag; caaactggtgtaccaaccaa; tgtaccgagttcaactgtat; ttgacgtgcctctgataaga; tagtagttgtctgattgtcc; aagtaaccttttgttggtgca; ccttggttgaatagtcttga; ctcatattgagttgatggct; gccaatttaaactcaccaga; tagagtcagcacacaaagcc; acttctgtgggaagtgtttc; agtagtatgtagccatactc; cacaggcgaactcatttact; atcatattaggtgcaagggc; aaggctttaagtttagctcc; ctgaattgtgacatgctgga; cccaaccgtctctaagaaac; tgcgggagaaaattgatcgt; cccctcttagtgtcaataaa; tctacccataaagccatcaa; agtagccaaatcagatgtga; cgcacagaattttgagcagt; tactgaatgccttcgagttc; gtgttccagttttcttgaaa; ggtaatcatcttcagtacca; gcttttagaggcatgagtag; catagggctgttcaagttga; ataggcacacttgttatggc; gtaattcagatactggttgc; aagccaatcaaggacgggtt; aagtgtctcaccactacgac.

Each probe was dissolved in water to a final concentration of 100 µM. The same set of probes was combined with equal volumes of each probe to get a stock of 100 µM mixed probes. The mixed probes were further desalted using ethanol precipitation. Briefly, 120 µL 100 µM probes were mixed with 12 µL 3 M sodium acetate (pH 5.5), followed by 400 µL ethanol. After precipitation at -80C overnight, probes were pelleted through centrifugation at 12,000 × g for 10 min at 4 °C, washed with precooled 70% (vol./vol.) ethanol three times, air dried, and dissolved in water to make a 100 µM solution of probes. Then, 18 µL 100 µM probes were mixed with 2 µL 1 M NaHCO₃ (pH 8.5), followed by 100 µg Alexa Fluor™ 647 succinimidyl ester (NHS) (Invitrogen, A37573) or CF568 succinimidyl ester (NHS) (Biotium, 92131) dissolved in 2 µL dry DMSO (Invitrogen, D12345). The mixture was incubated for 3 days at 37 °C in the dark for conjugation and purified for 3 rounds using Monarch PCR & DNA Cleanup Kit (5 µg) (NEB, T1030S) following the manufacturer's instructions. The estimated labeling efficiency of probes was calculated using Eq. (1):

$$\text{Modification ratio} = \frac{20}{(A_{\text{base}} \times \varepsilon_{\text{dye}})/(A_{\text{dye}} \times \varepsilon_{\text{base}})} \tag{1}$$

where $\varepsilon_{\text{dye}}$ is 239,000 cm⁻¹M⁻¹, $\varepsilon_{\text{base}}$ is 8,919 cm⁻¹M⁻¹, $A_{\text{base}}$ is the absorbance of the nucleic acid at 260 nm, and $A_{\text{dye}}$ is the absorbance of the dye at 650 nm. For the probes labeled with CF568, $\varepsilon_{\text{dye}}$ is 100,000 cm⁻¹M⁻¹, $\varepsilon_{\text{base}}$ is 8919 cm⁻¹M⁻¹, $A_{\text{base}}$ is the absorbance of the nucleic acid at 260 nm, and $A_{\text{dye}}$ is the absorbance of the dye at 562 nm.

## RNA FISH, immunofluorescence (IF), and CellMask staining

Fixed cells from BSL-3 as described above were washed twice with a freshly prepared 0.1% NaBH₄ solution at room temperature for 5 min, and washed with PBS three times. For staining without CellMask (Thermo Fisher, C10046), cells were permeabilized in 70% ethanol at 4 °C overnight. For CellMask staining, cells were permeabilized in 0.1% Triton X-100 at room temperature for 30 min.

For RNA FISH staining, permeabilized cells were washed with 200 μL Wash Buffer A [40 μL Stellaris RNA FISH Wash Buffer A (LGC Biosearch Technologies, SMF-WA1-60), 20 μL deionized formamide, 140 μL H2O] at room temperature for 5 min, and incubated with 110 μL Hybridization Buffer [99 μL Stellaris RNA FISH Hybridization Buffer (LGC Biosearch Technologies, SMF-HB1-10), 11 μL deionized formamide] containing 1.1 μL 12.5 μM vgRNA FISH probes for 4 h at 37 °C in the dark. Then cells were washed with Wash Buffer A for 30 min at 37 °C in the dark, washed with Wash Buffer A containing DAPI for 30 min at 37 °C in the dark, and stored in Wash Buffer B (LGC Biosearch Technologies, SMF-WB1-20) for imaging. DAPI was only added to the samples for confocal imaging and not added to the samples for SR imaging.

For IF staining with antibodies, permeabilized cells were washed with PBS twice, incubated with 3% BSA in PBS at room temperature for 30 min, and incubated with primary antibodies in PBS at 37 °C for 1 h. After incubation with primary antibodies, cells were washed twice with PBST buffer (0.1% Tween-20 in PBS) at room temperature for 5 min, washed with PBS once, incubated with secondary antibodies in PBS at room temperature for 30 min, washed with PBST buffer three times at room temperature for 5 min, and stored in PBS for imaging.

For simultaneous RNA FISH and IF staining, permeabilized cells were washed with 200 μL Wash Buffer A at room temperature for 5 min, and incubated with 110 μL Hybridization Buffer (99 μL Stellaris RNA FISH Hybridization Buffer, 11 μL deionized formamide) containing 1.1 μL 12.5 μM vgRNA FISH probes, 1 U/μL RNase inhibitor (NxGen, F83923-1), and primary antibodies for 4 h at 37 °C in the dark. Then cells were washed with 2xSSC buffer once, washed with Wash Buffer A containing secondary antibodies for 30 min at 37 °C in the dark, washed with Wash Buffer A for 30 min at 37 °C in the dark, washed with Wash Buffer B once, and stored in Wash Buffer B for imaging. For CellMask staining, several more steps were performed from here. Cells were washed with PBS once, stained with 1:20k CellMask and 1 U/μL RNase inhibitor in PBS for 20 min at room temperature in the dark, and washed with PBS three times before imaging.

## RNA FISH and IF staining of purified virions

8-well μ-slides were first treated with poly-D-lysine solution (Thermo Fisher, A3890401) at 4 °C overnight. Then in the BSL-3 facility, the poly-D-lysine solution was removed and 150 μL SARS-CoV-2 WA1 (P3) virus solution of titer 1.82·10⁵ PFU/mL was added into one well of poly-D-lysine-treated 8-well μ-slides for incubation at 4 °C for 24 h to coat the virions onto the surface of the well. After incubation, the medium containing virions was removed and the well was washed with PBS twice. Virions on the surface of the well were fixed with 4% PFA in PBS for 1 h at room temperature and the sample was removed from BSL-3. The sample was washed twice with a freshly prepared 0.1% NaBH₄ solution at room temperature for 5 min, and then washed with PBS three times. The fixed virions were permeabilized in 70% ethanol at 4 °C overnight and washed with PBS twice. For the group with Proteinase K digestion, virions were incubated with 0.2 mg/mL Proteinase K (NEB #P8107S) in 120 μL PBS at 37 °C for 30 min and washed with PBST buffer three times. Virions were washed with Wash Buffer A once and incubated with 110 μL Hybridization Buffer (99 μL Stellaris RNA FISH Hybridization Buffer, 11 μL deionized formamide) containing 1.1 μL 12.5 μM vgRNA FISH probes, 1 U/μL RNase inhibitor, and primary antibodies for 4 h at 37 °C in the dark. Then virions were washed with 2xSSC buffer once, washed with Wash Buffer A containing secondary antibodies for 30 min at 37 °C in the dark, washed with Wash Buffer A for 30 min at 37 °C in the dark, washed with Wash Buffer B once, and stored in Wash Buffer B for imaging.

## Labeling of newly replicated RNA with BrUTP

Vero E6 cells cultured in 8-well μ-slides were infected with SARS-CoV-2 WA 1 (USA212 WA1/2020) as described above. After 24 h of incubation, the culture medium was switched to low glucose DMEM medium (Thermo Fisher, 10567014) supplemented with 20 mM glucosamine for 30 mins to deplete uridine. Both infected and uninfected groups were further treated with 15 μM actinomycin D (Sigma, A4262) at 37 °C for 30 min to inhibit cellular transcription. To transfect cells with BrUTP, each well of cells was treated with 10 mM BrUTP (Sigma, B7166) and 12 μL Lipofectamine 2000 (Thermo Fisher, 11668030) for 1 h at 37 °C. Cells were then washed twice with PBS, followed by 4% PFA and 0.1% glutaraldehyde fixation for 1 h, and removed from BSL-3 following BSL-3 SOP of sample removal. The fixed cells were then washed twice with a freshly prepared 0.1% NaBH₄ solution at room temperature for 5 min, washed with PBS three times, permeabilized in 70% ethanol at 4 °C overnight, and washed twice with PBS.

The co-staining of BrU and nsp12 with antibodies follows the IF staining procedure detailed in the section "RNA FISH, immuno-fluorescence (IF), and CellMask staining". For the co-staining of BrU and vgRNA, cells were first incubated with BrU antibody and 1 U/μL RNase inhibitor in PBS at 37 °C for 30 min. Cells were then washed twice with PBST buffer, washed once with PBS, and incubated with the secondary antibody and 0.5 U/μL RNase inhibitor in PBS at room temperature for 30 min. Cells were then washed with PBST buffer three times, fixed again with 4% PFA and 0.1% glutaraldehyde in PBS at room temperature for 10 min, and washed with PBS three times. After that, cells were washed with Wash Buffer A at room temperature for 5 min, incubated with 110 μL Hybridization Buffer containing 1.1 μL 12.5 μM vgRNA FISH probes and 1 U/μL RNase inhibitor for 4 h at 37 °C in the dark. Then cells were washed twice with Wash Buffer A for 30 min at 37 °C in the dark and stored in Wash Buffer B for imaging.

## Spinning disk confocal microscopy

Confocal microscopy was performed at the Stanford University Cell Sciences Imaging Core Facility with a Nikon TiE inverted spinning disk confocal microscope (SDCM) equipped with a Photometrics Prime 95B camera, a CSU-X1 confocal scanner unit with microlenses, and 405 nm, 488 nm, 561 nm, and 642 nm lasers, using the 60×/1.27 NA PLAN APO IR water immersion objective. Images were taken using NIS Elements software version 4.60 with Z stacks at 0.3 μm steps. The camera pixel size of SDCM is 0.183 μm/pixel and the pinhole size is 50 μm. Only one Z slice is used for all images shown.

## Analysis of confocal data

To extract the intensity of vgRNA, dsRNA and RdRp in each infected cell (Supplementary Fig. S2c, e–h), the summation projection of each z stack was created by Fiji[62]. The intensity of each target species in each cell was measured by Fiji, subtracting the background of the same color channel. The infected cells were characterized manually into three types based on the morphology of vgRNA. Type 1 shows scattered dot-like localization of vgRNA. Type 3 shows large clustered vgRNA. Type 2 contains features of both type 1 and type 3.

## Optimization of antibody concentrations

We optimized the concentration of antibodies in this study by quantifying their signal-to-background ratio (SBR), where the signal is the brightness of the IF labeling in the cells that express the given target (virus-infected sample or cells expressing Sec61β-GFP), and the background is the brightness in the negative control cells (not-infected or WT cells).

To optimize the concentration of primary antibodies against the viral targets, different concentrations of the primary antibody were applied to stain Vero E6 cells in SARS-CoV-2-infected and not-infected samples under a constant secondary antibody concentration (Supplementary Fig. S19). To optimize the concentration of secondary antibodies, different concentrations of the secondary antibody were applied to stain Vero E6 cells in infected (virus+) and not-infected (virus−) samples under a constant primary antibody concentration (Supplementary Fig. S20). For each cell, an 11 pixel × 11 pixel box was drawn in the region with brightest signal in the cell and the mean intensity within that region was measured to represent the intensity of target antibody in that cell. The SBR was calculated, after subtraction of the dark signal $I_{dark}$, using Eq. (2):

$$SBR = \frac{<I_{virus+} - I_{dark}>}{<I_{virus-} - I_{dark}>} \qquad (2)$$

To optimize the concentration of the anti-GFP antibodies, different concentrations of primary antibody were applied to stain Vero E6 Sec61β-GFP cells and WT Vero E6 cells under a constant secondary antibody concentration (Supplementary Fig. S19). For each cell, an 11 pixel × 11 pixel box was drawn in the region with the brightest signal in the cell and the mean intensities of both the GFP and the antibody signals within that region were measured after subtraction of the dark signals. To account for the variable expression levels among different cells, the IF signal $I_{IF}$ was normalized by the GFP signal $I_{GFP}$ within the given region. The SBR was calculated using Eq. (3):

$$SBR = \frac{<I_{IF,Sec61\beta-GFP}/I_{GFP,Sec61\beta-GFP}>}{<I_{IF,WT}/I_{GFP,WT}>} \qquad (3)$$

For the primary antibodies against GFP, nsp3, nucleocapsid, nsp12 and for the secondary antibody for the dsRNA labeling, we chose the antibody concentration that produces the highest SBR as the optimal concentration. For the primary antibodies against spike S2 and dsRNA and for the secondary antibody for the spike S2 labeling, we chose the concentration that yields the second highest SBR because it provides a significantly lower non-specific background with only a minor decrease of the estimated SBR.

**Optical setup for SR microscopy**
(d)STORM SR microscopy was performed on a custom-built system (Supplementary Fig. S21), consisting of a Nikon Diaphot 200 inverted microscope frame with an oil-immersion objective 60x/1.35 NA (Olympus UPLSAPO60XO) and a Si EMCCD camera (Andor iXon Ultra 897). We used 642 nm and 560 nm 1 W continuous-wave (CW) lasers (MPB Communications Inc.) for excitation of AF647 or CellMask and CF568 or CF583R, accordingly. For reactivation of fluorophores from the dark state we used a 405 nm 50 mW CW diode laser (Coherent OBIS). All laser beams were expanded and co-aligned in free space and coupled into a square-core multi-mode fiber with a shaker for speckle reduction (Newport F-DS-ASQR200-FC/PC). The output tip of the fiber (200 × 200 μm² core size) was imaged with a 10x/0.25 NA objective and magnified to achieve a square illumination region of 47.6 × 47.6 μm² with a constant intensity in the sample image plane of the main objective. The fluorescence was split from the excitation light with a multi-band dichroic mirror (ZT405/488/561/640rpcv2, Chroma) and filtered with dichroic filters (ZET635NF, ZET561NF, T690LPxxr, all Chroma). The fluorescence of AF647 and CellMask was additionally filtered with a band-pass filter (ET685/70 M, Chroma) and that of CF568 and CF583R with a combination of 561LP and 607/70BP (Semrock, EdgeBasic and BrightLine). The sample image was focused with a tube lens ($f$ = 400 mm) on the EMCCD camera, providing a pixel size of 117 × 117 nm² in sample coordinates.

Axial drift was compensated with a custom Focus Lock system[63]. We used an 808 nm fiber-coupled diode laser (Thorlabs S1FC808) whose output fiber tip was conjugated with the back focal plane of the imaging objective, allowing changing the angle of this beam out of the objective by translating the fiber tip (Supplementary Fig. S21). This inclined beam was partially reflected from the coverslip-water interface and the reflected beam was focused with a cylindrical lens onto a CMOS sensor (UI-3240CP-NIR, IDS Imaging). The 808 nm beam was aligned such that the image of the reflected beam would shift laterally when the axial position of the sample changes. The sample was mounted on two stacked piezo stages (U-780.DOS for coarse and P-545.3C8S for fine movement, both Physik Instrumente). The position of the reflected beam image was recorded when the sample was set at the desired Z position for imaging. During imaging, the Z-position of the fine stage was directed to move proportionally to the shift of the reflected beam image from the recorded position, compensating for Z-drift. The Focus Lock control code was programmed in Matlab (MathWorks, Inc.).

**SR imaging procedure**
For (d)STORM, the sample chamber was filled with 300 μl of a photoblinking buffer consisting of 200 U/ml glucose oxidase, 1000 U/ml catalase, 10% w/v glucose, 200 mM Tris-HCl pH 8.0, 15 mM NaCl and 50 mM cysteamine. The buffer was prepared using the following stock solutions[48]: 1) 4 kU/ml glucose oxidase (G2133, Sigma), 20 kU/ml catalase (C1345, Sigma), 25 mM KCl (P217, Fisher), 4 mM TCEP (646547, Sigma), 50% v/v glycerol (BP229, Fisher) and 22 mM Tris-HCl pH 7.0 (BP1756, Fisher), stored at −20 °C; 2) 1 M cysteamine-HCl (30080, Sigma), stored at −20 °C; 3) 37% w/v glucose (49139, Sigma) with 56 mM NaCl (S271, Fisher) and 0.74 M Tris-HCl pH 8.0 (J22638.AE, Fisher), stored at +4 °C. For samples with RNA FISH labeling, the buffer was supplemented with 1 U/μl of an RNase inhibitor (302811, LGC Biosearch Technologies).

The SR imaging started with a DL image of cells from each fluorophore at a low power (e.g., 2 W/cm²). For (d)STORM acquisitions, we began with AF647 or CellMask, followed by CF568 or CF583R[64]. We used an excitation power density of ~20 kW/cm² for shelving and blinking of CF568, ~13 kW/cm² for CF583R and 6–20 kW/cm² for AF647. The power density of the 405 nm illumination for both dyes was increased from 0 to 50 W/cm² throughout an acquisition to keep the reactivation rate approximately constant. The exposure time was 10.57 ms per frame and the calibrated EM gain was either 43 or 84. The image recording started after the initial shelving phase upon observation of clear SM blinking; the blinking movies were acquired for approximately $6\cdot10^4$–$8\cdot10^4$ frames for each fluorophore.

**SR data analysis**
SM movies were processed with the ThunderStorm plugin[65] for Fiji. First, the images were filtered with a wavelet filter with a b-spline order of 3 and a scale of 2. The coarse localizations were found as local maxima with an 8-neighborhood connectivity and a threshold of 2·std(Wave.F1). These localizations were weighted least squares-fitted with the integrated Gaussian model using a radius of 4 pixels and an initial sigma of 1.1. Then, we performed drift correction estimated by cross-correlation between successive subsets of localizations in ThunderStorm, or in SharpViSu[66] when the drift correction in ThunderStorm was unsuccessful. For further processing, we kept only localizations with fitted sigma between 160 nm and 80 nm. This choice effectively rejects molecules away from the focal plane, providing an approximate axial sectioning of the images to roughly 500 nm[67].

For image registration, we imaged 200 nm TetraSpeck beads (T7280, Thermo Fisher Scientific) in both channels, whose images were processed similarly to the SM movies. The transformation between the channels was calculated using an affine transformation with help of Matlab function 'fitgeotrans'. The calculated transformation was then

applied to the CF568 or CF583R localizations using a Matlab function 'transformPointsInverse'.

Localizations found within 50 nm on consecutive frames that could originate from multiple localizations of a single molecule were treated in two ways. For SR images, to improve the resolution, these localizations were refined by selecting them from a normal distribution with a mean at the weighted mean of the initial localizations and a standard deviation (SD) that equals $120 \cdot (N_{ph})^{-1/2}$ nm, where $N_{ph}$ is the total number of photons acquired from all localizations in the given consecutive series[48]. For data analysis other than SR image reconstruction, to suppress overcounting, the localizations of the consecutive series were reduced to a single localization at the weighted mean position. The weights of localizations were proportional to the photon counts of these individual localizations. After this correction, the SR data of antibody-detected Spike, N, nsp12, BrU, nsp8, nsp7 was additionally filtered by removing localizations that had 3 or less neighbors within 30 nm. SR images were reconstructed as 2D histograms with a bin size of $20 \times 20$ nm². However, SR images where one of the channels contained the CellMask labeling had a bin size of $30 \times 30$ nm². SR images acquired with CellMask were additionally filtered with a Gaussian filter with $\sigma = 0.5$ pixels.

### Cluster analysis with BIC-GMM

Gaussian Mixture Models (GMM) implemented in Python were fitted to vgRNA and dsRNA localization datasets, yielding a representation of localization densities as a collection of potentially elliptical and/or rotated 2D Gaussians. The number of components most suitable for each field of view was determined using an iterative grid search, evaluating 4 candidate GMMs using the Bayesian Information Criterion (BIC)[68]. The first grid iteration tested [1, 2500] components with test points $t_i = \{1, 834, 1667, 2500\}$, where $i$ denotes the index in the set such that $t_0 = 1$. For each iteration of the grid search, the model with the lowest BIC (corresponding to the best candidate), $t_k$ was selected, and the next iteration of the grid was narrowed, to be bounded by [$t_{\max (k-1, 0)} + 1$, $t_{\min (k+1, 3)} - 1$], until the stride of the grid was 1 component, or the test point with the best BIC was on a rail ($k = 0$ or 3). To reduce memory requirements, this GMM optimization was performed on a random subset of up to 200,000 localizations from each data set, but the optimized GMM was then used to predict a component assignment for all original localizations. These components were regarded as clusters, and refined by removing localizations with a log probability of being an event from their assigned Gaussian component of less than −25. The radius of gyration, Rg, was then calculated for each cluster, and the number of localizations in each cluster, $N_{loc}$, was used to approximate a cluster density as $\delta = N_{loc}/(\pi \cdot Rg^2)$. Clusters with $\delta$ below a threshold of 0.008 localizations/nm² for dsDNA, or below an ROI-dependent threshold between 0.005 and 0.013 localizations/nm² for vgRNA, were removed from further quantification as sparse background. This analysis and resulting visualizations were carried out in the PYthon Microscopy Environment (https://doi.org/10.5281/zenodo.4289803)[69], using a plugin, bic-gmm[70] that leverages the scikit-learn GMM implementation[71].

### Counting of vgRNA molecules in the clusters

The number of vgRNA molecules in a vgRNA cluster was defined as a quotient between the number of vgRNA-FISH localizations in the cluster and the average number of localizations produced by a single FISH-labeled vgRNA molecule in the given cell. The average number of localizations per vgRNA molecule was estimated from isolated nanoscale vgRNA puncta in the cytoplasm (Supplementary Fig. S3a). This number was defined as the median of the number of localizations within 50 nm from each localization in the region with vgRNA puncta. The estimated number of vgRNA molecules was calculated for every cluster determined by the BIC-GMM cluster analysis and the median value per cell was shown in a chart (Supplementary Fig. S3b, c).

### Counting of nsp12 puncta in the vgRNA clusters

The center of nsp12 puncta is obtained by fitting the SR images in ThunderStorm[65]. The SR localizations of nsp12 were first converted into a 2D histogram image with a bin size of $20 \times 20$ nm². The approximate localization of the center was found as a centroid of connected components with a threshold of 5·std(Wave.F1) without filter. These localizations were least squares-fitted with the integrated Gaussian model using a fitting radius of 2 pixels and an initial sigma of 0.4. We next removed duplicates among localizations within a 20 nm radius. The puncta whose sigma were smaller than 5 nm were further filtered out to avoid localizing single-pixel-sized background localizations. For each vgRNA cluster with its center and the radius of gyration (Rg) determined using BIC-GMM, we counted the number of nsp12 puncta within a 1.5·Rg distance of the center of the vgRNA cluster. For nsp12 puncta found within the cutoff distance of more than one vgRNA cluster, we assigned them to their closest cluster based on the relative distance $d$/Rg, with $d$ being the distance between the center of the vgRNA cluster and center of the nsp12 punctum.

### Bivariate pair-correlation functions

For calculation of bivariate pair-correlation functions[29] $g_{12}(r)$, we first manually selected the cytoplasmic regions with dense vgRNA clusters. The pair-correlation functions were calculated by counting the number of localizations of the second species within a distance between $r$ and $r + dr$ from each localization of the first species. These were normalized by dividing the number of localizations by the area of the corresponding ring of radii $r$ and $r + dr$ and by the average density of the second species in the region. Finally, the obtained numbers were averaged across the localizations of the first species. $r$ was scanned over the range between 0 and 500 nm and $dr$ was set to 1 nm. For the complete spatial randomness (CSR) case, a test CSR dataset was generated with the same average density as for the experimental case across the same ROI. $g_{12}(r)$ traces were calculated from these CSR datasets as described above. No edge effect correction was performed leading to a slight decrease of $g_{12}(r)$ at large $r$. Plots in the figures display experimental and CSR $g_{12}(r)$ for each analyzed cell as faint lines as well as the mean $g_{12}(r)$ calculated from all cells in bold lines.

### Estimation of RNA FISH labeling efficiency in virions

Dye molecules inside virions were counted using fluorescence bleaching with SM calibration. Virions attached to the coverslip were labeled using the RNA-FISH + IF protocol with PFA-only fixation. The density of virions was around $0.5 \, \mu m^{-2}$ ensuring observation of most virions as single DL spots without overlap (Supplementary Fig. S1a, d). vgRNA was FISH-labeled with AF647 and spike protein was IF-stained with CF568. Glass-bottom chambers with virions were kept in PBS for this experiment. Samples were illuminated with 642 nm light at 20 W/cm² and were imaged with an exposure time of 200 ms and an EM gain of 43 until bleaching of all AF647 in the imaging region (around 200 s). A separate DL image of spike was taken with 560 nm excitation. The AF647 bleaching movies were processed in ThunderStorm using a wavelet filter with a b-spline order of 3 and a scale of 2, a local maximum approximate localization with a threshold of 1.2·std(Wave.F1) and an 8-neighborhood connectivity. These localizations were weighted least squares-fitted with the integrated Gaussian model using a radius of 3 pixels and an initial sigma of 1.1. Then, we kept only localizations with $sigma < 160$ nm & $sigma > 80$ nm and removed duplicates within 300 nm on each frame.

Further processing was done in Matlab with a custom script. We considered only vgRNA-AF647 localizations that had a spike-CF568 signal within 200 nm to avoid counting AF647 molecules outside virions. The bleaching time traces (Supplementary Fig. S1c, f) were found by searching in consecutive frames within 200 nm of the localization from the first frame and allowing up to 5 empty frames between frames with detections. The number of bleaching steps was defined as the

rounded quotient between the initial and the final brightness of a spot in a time trace serving as the SM calibration. For each bleaching trace, the initial brightness (in photons) was defined as the median value of the brightness in the first 4 localizations and the final brightness as the median brightness value of the last 4 localizations. If the trace contained only 7-8 detections, the range for the initial and the final brightness was reduced to 3 frames; for traces with 5-6 detections, this was reduced to 2; for traces with 3-4 frames – to 1; for traces containing only 1 or 2 detections, the number of bleaching steps was set to 1. For each analyzed region containing around 200 bleaching traces, the number of bleaching steps was fitted with a zero-truncated Poisson distribution (Supplementary Fig. S1g, h). The expected values ± SD obtained from the fit of 5 regions for each of not-treated and PK-treated cells are shown in a chart (Supplementary Fig. S1i).

## Statistics and reproducibility

The experimental measurements were replicated 3 times (4 times for vgRNA) at 24 hpi and 2 times at 6 hpi, starting with cell growth and infection at the BSL-3 facility (independent biological replicates). All biological replicates were successful. Each sample preparation resulted in approximately 10 wells with different target pairs. For each prepared sample well (each label pair), typically 5–10 cells were imaged using SR microscopy and 50–100 cells were imaged using confocal microscopy. Because different targets appear in many pairs of labels, each target was imaged multiple times. The total number of infected cells imaged in SR microscopy with particular targets was: vgRNA, >150 cells at 24 hpi, >50 cells at 6 hpi; dsRNA, >100 cells at 24 hpi, >30 cells at 6 hpi; nsp12, >40 cells at 24 hpi, >20 cells at 6 hpi; Sec61β, >40 cells at 24 hpi, >25 cells at 6 hpi; nsp3, >40 cells at 24 hpi, >30 cells at 6 hpi; spike protein, >40 cells at 24 hpi, >25 cells at 6 hpi; nucleocapsid protein, >30 cells at 24 hpi, >20 cells at 6 hpi. All $p$-values were obtained using an unpaired two-sample two-tailed Student's $t$ test. Blinding was not possible in this study, because due to intrinsically low throughput of SR microscopy, only cells with a positive signal of SARS-CoV-2 markers were selected to be imaged and analyzed in the infected cell groups.

## Reporting summary

Further information on research design is available in the Nature Portfolio Reporting Summary linked to this article.

## Data availability

The data for analysis figures generated in this study are provided in the Source Data file. Source data are provided with this paper.

## Code availability

The code utilized in this study is available at Stanford Digital Repository at https://doi.org/10.25740/td954gx5320.

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

## Acknowledgements

We thank Amol Pohane for assistance with the cell culturing in the BSL-3 facility. We thank Leiping Zeng for sample preparation and discussion about the results and experimental plan with the other authors. This work was supported in part by the National Institute of General Medical

Sciences Grant Nos. R35GM118067 (to W.E.M.) and the National Institutes of Health Common Fund 4D Nucleome Program No. U01 DK127405 (to L.S.Q.). We also acknowledge Stanford University Cell Sciences Imaging Core Facility (RRID:SCR_017787). M.H. and Y.Z. acknowledge support by the Stanford School of Medicine Dean's Postdoctoral Fellowship. L.S.Q. is a Chan Zuckerberg Biohub – San Francisco Investigator, and W.E.M. is a Sarafan ChEM-H Fellow.

## Author contributions

L.A., M.H., L.S.Q. and W.E.M. conceived the project. L.A. designed the optical set-up, performed the SR acquisitions and data analysis. M.H. performed cell culture, labeling and confocal imaging. Y.Z. performed confocal and SR data analysis and helped with sample preparation and confocal imaging. J.G. and P.P. performed SARS-CoV-2 infection experiments at the BSL-3 facility with staff listed in the Acknowledgements. A.R.R. contributed to the concept and SR experiments at the early stages of the project. A.E.S.B. designed the BIC-GMM cluster analysis method and contributed to the optical set-up design. A.B. performed portions of the BrU SR imaging. L.A. and W.E.M. wrote the manuscript with input from all authors.

## Competing interests

The authors declare no competing interests.
