## [Peer Review File · Nature Communications]

Nanoscale cellular organization of viral RNA and proteins in SARS-CoV-2 replication organellesREVIEWER COMMENTS

Reviewer #1 (Remarks to the Author):

In this manuscript by Andronov et al., the investigators used super-resolution (SR) microscopy and, in conjunction with conventional confocal microscopy, detailed the organization of cellular structures that are thought to be involved in SARS-CoV-2 replication. The major claim of this study is that the replication organelles supporting virus replication at early (6 hpi) and late (24 hpi) infection are rather distinct. Specifically, at the early time point, the viral genomic RNA (vgRNA) appears as scattered clusters/puncta throughout the cytoplasm, but at the later time point, the vgRNA are concentrated in perinuclear dense/large structures. Overall, although there are some potentially exciting points suggested by this study, the investigators need to perform additional experiments to fully justify the major claim.

Major points:

1. Because this study is largely focused on elucidating virus replication events, using the dsRNA marker makes sense because it marks the active replication event. However, the group appears to continue to track the localization pattern of the viral genomic RNA (vgRNA). vgRNA has three different fates: it can be used 1) for translation of the non-structural viral proteins, 2) as templates for further rounds of replication, and 3) during packaging to assemble the new viral progeny. Thus, tracking the vgRNA pattern does not necessarily track the replication event.
2. While the SARS-CoV-2 replication structures at 6 hpi are consistent with the DMVs reported by many groups, the dense and large (approximately 1 micron) perinuclear structures at 24 hpi – postulated to be sites of virus replication – are unusual and potentially novel. Are these structures really membranous structures (i.e. is replication occurring within a membrane-bound structure?). I would ask the investigators to perform a standard TEM analysis to address this question. If replication at this later time point is happening within a membranous structure, the vgRNA should be protected against RNase digestion, (in the context of a cell-based semi-permeabilized system).
3. As indicated above, presence of vgRNA does not definitively indicate a replication site, as the vgRNA can be used for packaging to make new viral progenies. Along this logic, can the investigators rule out the possibility that the large/dense perinuclear structures at 24 hpi are not virus assembly sites? Staining for the presence of structural proteins (e.g. Spike or M protein) at these sites would be one reasonably straightforward approach to address this question.
4. Our lab are not experts in SR microscopy. However, it is a bit confusing that while the postulated replication structures at 6 vs 24 hpi in Figure 3 appear different by confocal microscopy, they seem similar by SR microscopy. Please explain.

Minor points:

1. Please provide more background regarding the SARS-CoV-2 infection cycle – this will give the audience a sense of where the replication event occurs.
2. In Figures 1,2, 3, and 5, please do not put the inset in the image – this is rather disruptive in understanding the entire image.

Reviewer #2 (Remarks to the Author):

Andronov et al. present a novel study that sheds light on the cellular organization of viral RNA and proteins during SARS-CoV-2 replication in host cells. High resolution microscopy (dSTORM) was combined with immunofluorescence and smFISH to elucidate the positioning and organization of SARS-CoV-2 RNAs, proteins, and cellular membranes during infection. Novel observations include the finding that replication organelles are organized differently at early and late stages of infection, as well as a striking rearrangement of nsp3 to the periphery of replication organelles during later stages of infection and the simultaneous observation of both vgRNA and dsRNA within the same DMVs.

The experiments reported in the paper are generally well designed and suitable quantitative analysis is performed. Overall, this is a nicely presented study in which the results support the majority of the important conclusions of the work, although I do have a small number of queries and concerns (detailed below). The results should be of significant and wide interest for the journal's readership, and may be relevant for therapeutic strategies against SARS-CoV-2.

Major:

- 1) It has previously been shown that anti-dsRNA antibodies like rJ2 bind quite poorly to dsRNA and may underestimate SARS-CoV-2 replication (only 10% of infected cells stained positive at 6 hpi) (Lee et al., eLife, 2022). Is it possible that dsRNA labelling is being vastly underestimated in your experiments and would this have any consequences for your results? Is it possible to use smFISH probes designed against the negative sense vgRNA strand as a more efficient way of detecting dsRNA (assuming that the negative sense RNA is representative of the location of dsRNA molecules).
- 2) I find it quite strange that overall nsp12 numbers don't increase between 6 and 24 hours post infection. One possibility that I would like to see addressed if possible is that the nsp12 antibody used in the study isn't actually binding to the actively replicating form of the protein (which would in fact be part of a large complex of nsps and so assumed to be quite inaccessible). If this is the case it may explain why there isn't a large increase in nsp12 molecules detected.

Minor:

- 1) It may just be a personal preference, but I found it a little hard to follow the order of the figures, particularly where the graphs are contained within other images. Whilst it may not be the most efficient use of space it may aid the reader to have the various panels presented in the order that they are described in the text, and with the graphs placed separately alongside the images.
- 2) Fig S1 J, is vgRNA also visible inside spike labelled virions at the cell periphery?
- 3) Page 7, line 148. Can you speculate why the observed circular structures that the vgRNA forms may be hollow? Might this support a model whereby vgRNA is copied by polymerase located at the DMV membrane, i.e. is this suggesting that the viral polymerase may be membrane-associated?
- 4) Figure S3 A, please could you indicate on the images some examples of which clusters are taken to be single vgRNA punctum (those that are used as calibration)?
- 5) Page 11, line 220. Fig 3g is referenced in the text before e and f, which is hard to follow.
- 6) Page 12, Figure 3h seems a little out of place. It isn't mentioned in the text until much later, making it difficult to understand the relevance of this panel in figure 3, and so I would suggest moving this panel to Figure 4.

Response to Reviewer #1 (Remarks to the Author):

For convenience, we show the referee comments in *italics*, and our replies in blue. Textual changes made in the manuscript are shown in red.

In this manuscript by Andronov et al., the investigators used super-resolution (SR) microscopy and, in conjunction with conventional confocal microscopy, detailed the organization of cellular structures that are thought to be involved in SARS-CoV-2 replication. The major claim of this study is that the replication organelles supporting virus replication at early (6 hpi) and late (24 hpi) infection are rather distinct. Specifically, at the early time point, the viral genomic RNA (vgRNA) appears as scattered clusters/puncta throughout the cytoplasm, but at the later time point, the vgRNA are concentrated in perinuclear dense/large structures. Overall, although there are some potentially exciting points suggested by this study, the investigators need to perform additional experiments to fully justify the major claim.

We thank the reviewer the time spent reviewing our manuscript, for the feedback and for the suggestions, which have helped to improve the manuscript.

Major points:

1. Because this study is largely focused on elucidating virus replication events, using the dsRNA marker makes sense because it marks the active replication event. However, the group appears to continue to track the localization pattern of the viral genomic RNA (vgRNA). vgRNA has three different fates: it can be used 1) for translation of the non-structural viral proteins, 2) as templates for further rounds of replication, and 3) during packaging to assemble the new viral progeny. Thus, tracking the vgRNA pattern does not necessarily track the replication event.

We appreciate this suggestion to clarify the focus of our work. We have focused on fate 2) because we found other replication players close to the perinuclear vgRNA clusters (dsRNA, RdRp, nsp3, and modified ER encircling them). To go further to underscore that the vgRNA clusters are involved in replication, we have now imaged newly replicated viral RNAs using metabolic labeling with BrUTP following Heinrich et al. 2010 (Ref. 35) and Gosert et al. 2002 (Ref. 34), and localized newly synthesized viral RNAs within the vgRNA clusters. The manuscript has been updated on p.13 as follows:

Finally, to confirm that the vgRNA clusters we observe contain newly replicated viral RNA, we provided brominated uridine (BrU) to the infected cells in the form of 5-bromouridine 5'-triphosphate (BrUTP) for 1 hour before fixation while endogenous transcription was inhibited by actinomycin D^{34,35}. Immunofluorescent labeling of BrU then highlights newly replicated RNA. Confocal and SR

imaging localizes RNA containing BrU to the perinuclear clusters of vgRNA (Fig. 3g, Supplementary Fig. S7) and close to nsp12 (Supplementary Fig. S8), further proving that these structures are the sites of active replication and transcription of viral RNA.

2. While the SARS-CoV-2 replication structures at 6 hpi are consistent with the DMVs reported by many groups, the dense and large (approximately 1 micron) perinuclear structures at 24 hpi – postulated to be sites of virus replication – are unusual and potentially novel. Are these structures really membranous structures (i.e. is replication occurring within a membrane-bound structure?). I would ask the investigators to perform a standard TEM analysis to address this question. If replication at this later time point is happening within a membranous structure, the vgRNA should be protected against RNase digestion, (in the context of a cell-based semi-permeabilized system).

Indeed, these are valid points, but we respectfully believe that we have addressed them. We imaged two membrane proteins (nsp3, Sec61b) and a general membrane label (CellMask) and all these labels are found encapsulating the large vgRNA structures that also contain nsp12, nsp7, nsp8, dsRNA and newly synthesized RNA (BrU) indicating viral replication. Fig. 4b and Fig. S9-S15 show multiple examples. Encapsulation of the vgRNA structures by membrane proteins and by CellMask labelling indicates that the vgRNA clusters are membrane-constrained structures. The close association of dsRNA with vgRNA clusters also suggests that these structures are inside DMVs because it is commonly accepted that dsRNA is found inside DMVs. Moreover, the size and the perinuclear localization of vgRNA clusters strongly resemble those of DMVs and of vesicle packets at later time points as reported by EM.

We agree that standard TEM analysis can localize membranes, and we have already referenced multiple prior efforts of this type (Wolff et al, 2020 [Ref. 25], Klein et al, 2020 [Ref. 5], Eymieux et al, 2021 [Ref. 22]). However, this method does not provide information about localization of specific RNAs or other replication markers. Immuno-TEM does not preserve membrane structure well. Resolving these questions with EM would require new technical developments (e.g., correlative light+cryo-electron microscopy) that are beyond the scope of the current study.

3. As indicated above, presence of vgRNA does not definitively indicate a replication site, as the vgRNA can be used for packaging to make new viral progenies. Along this logic, can the investigators rule out the possibility that the large/dense perinuclear structures at 24 hpi are not virus assembly sites? Staining for the presence of structural proteins (e.g. Spike or M protein) at these sites would be one reasonably straightforward approach to address this question.

This is a very helpful suggestion and we have performed new experiments to address this. We now imaged the structural proteins spike and nucleocapsid and found that both structural proteins have bivariate pair correlation functions show anti-correlation with the perinuclear vgRNA structures. On the other hand, small amounts of spike and nucleocapsid proteins can be found surrounding vgRNA

clusters, likely at the membrane of ROs. Of course, we find a great deal of evidence for virion formation at the periphery of the cell. Taken together, these results rule out the assembly of virions within the dense perinuclear vgRNA structures but suggest that early stages of virion assembly may start at the DMV/RO membranes once vgRNA leaves the ROs. Please see new Supplementary Fig S17, S18. The manuscript has been updated on p.18-19 as follows:

To search for a possible role of perinuclear vgRNA clusters in virion assembly, we co-imaged vgRNA with two SARS-CoV-2 structural proteins, spike and nucleocapsid (Supplementary Fig. S17, S18). Spike labeling forms typical ~ 150 nm hollow particles at the cell periphery, and we detect weak vgRNA signal in the center of some of these particles (Supplementary Fig. S17b), consistent with the structure of SARS-CoV-2 virions that contain a single vgRNA molecule. Inside the host cells, spike localizes at the nuclear envelope and in some cytoplasmic organelles; however, it is mostly excluded from the perinuclear vgRNA clusters (Supplementary Fig. S17a, c). Nucleocapsid protein demonstrates rather diffuse localization throughout the cytoplasm, in accordance with its function in the formation of SARS-CoV-2 ribonucleocapsid complexes⁴³, but is also excluded from the RO interior (Supplementary Fig. S18a). Nevertheless, in the perinuclear region we detect sparse localizations of both spike and nucleocapsid proteins next to the vgRNA clusters and between them, likely at the DMV membranes, as highlighted by anti-correlation of these proteins with vgRNA at $r < 200$ nm (Supplementary Fig. S17c, S18b), similar to the nsp3/vgRNA and Sec61 β /vgRNA pairs. The localization of nucleocapsid protein at the RO membranes has already been reported⁴⁴, and spike protein has a transmembrane domain⁴⁵ and tends to localize not only to virion membranes, but also to intracellular membranes, such as the nuclear envelope (Supplementary Fig. S17a); therefore, small amounts of spike can also be present at RO membranes. Our SR data suggests that while the vgRNA clusters are not directly involved in SARS-CoV-2 virion assembly, it is possible that early stages of virion assembly start at the RO membrane, once vgRNA molecules leave the ROs.

4. Our lab are not experts in SR microscopy. However, it is a bit confusing that while the postulated replication structures at 6 vs 24 hpi in Figure 3 appear different by confocal microscopy, they seem similar by SR microscopy. Please explain.

We certainly want to avoid this confusion. We have now replaced the SR image of a 6 hpi cell (Fig. 3c) with one that better reflects the typical type 1 infected cells at 6 hpi (sparse vgRNA clusters) and is more different from 24 hpi. To address this, the following explanation about the difference of SR vs DL images has been added on p11-12.

From comparison of DL and SR images, one may infer fundamentally different (large-scale) nsp12 structures at 6 hpi and 24 hpi in confocal microscopy (Fig. 3a-b). In DL microscopy, ROs do look like individual diffraction-limited dots at 6 hpi when they are sparse (Fig. 3a), *i.e.*, the average distance between them is larger than the diffraction limit (even though the individual RdRp complexes inside

ROs are still not resolved). The same organelles when they are dense at 24 hpi resemble large irregular blobs because the distance between the individual organelles becomes smaller than the diffraction limit (Fig. 3b). This filling in with optically overlapping ROs creates a misleading perception of distinct structures in confocal microscopy. However, SR microscopy, which sees spatial details on the scale of 20-40 nm, resolves both types of structures much better. The nsp12 puncta are small in both cases because they arise from individual RdRp enzymes, yet the vgRNA clusters are smaller at 6 hpi and larger at 24 hpi, which is a better representation of the size of these assemblies.

Minor points:

1. Please provide more background regarding the SARS-CoV-2 infection cycle – this will give the audience a sense of where the replication event occurs.

We agree that a more detailed background of SARS-CoV-2 infection cycle was needed. We have added an overview of SARS-CoV-2 infection cycle to the introduction (p4):

The SARS-CoV-2 life cycle starts with viral entry into a host cell, facilitated by binding of viral spike protein to its canonical receptor at the cell surface, the angiotensin-converting enzyme 2 (ACE2)¹⁶, or one of the alternative receptors¹⁷. The subsequent fusion of the viral and the host cell membranes releases the viral genetic material, positive-sense single-stranded viral genomic RNA (vgRNA), into the cytoplasm, where it is readily translated by host ribosomes. SARS-CoV-2 vgRNA (Fig. 1a) encodes at least 29 proteins, including structural proteins that make up the virions, and non-structural (NSPs) and accessory proteins that exist only within host cells and regulate various processes in the intracellular viral life cycle. All NSPs originate from polyproteins that are translated directly from vgRNA and are self-cleaved by viral proteases. Structural and accessory proteins are translated from shorter viral genome fragments called subgenomic RNAs (sgRNAs) that are transcribed from vgRNA.

Replication and transcription of the viral genome is carried out by the RNA-dependent RNA polymerase complex (RdRp), which is assembled from nsp12 (RdRp catalytic subunit) along with nsp7 and nsp8 (accessory subunits)¹⁸. RdRp first synthesizes either a full-length negative-sense copy of vgRNA or a subgenomic negative-sense copy of vgRNA, producing double-stranded RNA (dsRNA) that forms between vgRNA and the negative-sense copy. Next, using this negative-sense template, a new vgRNA or an sgRNA is generated by the same polymerase enzyme. Additional NSPs modify newly synthesized viral RNAs to form 5' cap structures¹⁹ that mimic cellular mRNAs to be translated by host ribosomes. The replication intermediates, such as dsRNA and uncapped RNAs, might be degraded or trigger innate immune response²⁰ and therefore need to be protected from cellular machinery. SARS-CoV-2 transforms host ER into DMVs²¹ that are abundant in the perinuclear region of infected cells^{4,5,22} and likely encapsulate dsRNA^{3,5} and newly synthesized viral RNA^{4,23}. However, the precise

intracellular localization of replicating RdRp enzymes and therefore of the replication events is not well established to date^{3,23,24}.

2. In Figures 1,2, 3, and 5, please do not put the inset in the image – this is rather disruptive in understanding the entire image.

We thank the referee for this suggestion since it may improve the understanding of the figures. It is a pity to waste the huge nuclear dark space, but we have now taken this suggestion. We have reformatted the figures and placed the insets outside the cell images wherever possible.

Response to Reviewer #2 (Remarks to the Author):

For convenience, we show the referee comments in *italics*, and our replies in blue. Textual changes made in the manuscript are shown in red.

Andronov et al. present a novel study that sheds light on the cellular organization of viral RNA and proteins during SARS-CoV-2 replication in host cells. High resolution microscopy (dSTORM) was combined with immunofluorescence and smFISH to elucidate the positioning and organization of SARS-CoV-2 RNAs, proteins, and cellular membranes during infection. Novel observations include the finding that replication organelles are organized differently at early and late stages of infection, as well as a striking rearrangement of nsp3 to the periphery of replication organelles during later stages of infection and the simultaneous observation of both vgRNA and dsRNA within the same DMVs.

The experiments reported in the paper are generally well designed and suitable quantitative analysis is performed. Overall, this is a nicely presented study in which the results support the majority of the important conclusions of the work, although I do have a small number of queries and concerns (detailed below). The results should be of significant and wide interest for the journal's readership, and may be relevant for therapeutic strategies against SARS-CoV-2.

We thank the reviewer for the time spent reviewing our manuscript, for the positive feedback and for the useful suggestions.

Major:

1) It has previously been shown that anti-dsRNA antibodies like rJ2 bind quite poorly to dsRNA and may underestimate SARS-CoV-2 replication (only 10% of infected cells stained positive at 6 hpi) (Lee et al., eLife, 2022). Is it possible that dsRNA labelling is being vastly underestimated in your experiments and would this have any consequences for your results? Is it possible to use smFISH probes designed against the negative sense vgRNA strand as a more efficient way of detecting dsRNA (assuming that the negative sense RNA is representative of the location of dsRNA molecules).

We understand the concern about the anti-dsRNA antibody. However, in our hands, there was no evidence of dsRNA underestimation with the J2 antibody. We reliably detected dsRNA that colocalizes with weak vgRNA labelling in early infection and we did not observe any cells with detectable vgRNA signal that did not have dsRNA labelling. To illustrate this, we have added Fig. S2d depicting an early infection cell with weak vgRNA signal that has clear colocalizing dsRNA labelling. Lee et al. used the J2 anti-dsRNA antibody in 1:500 dilution (2µg/ml), while we use it at 1:200 (5µg/ml). This provides 2-3x higher signal and signal to background ratio (Fig. S19) and therefore

detects a higher percentage of dsRNA molecules. There might be other experimental conditions that contributed to the difference in immunofluorescence efficiency. The manuscript has been updated on p. 8-9 as follows:

In confocal microscopy, dsRNA labeling was present in all cells with detectable vgRNA FISH fluorescence, including in early infection, demonstrating the high sensitivity of our dsRNA immunofluorescence detection (Supplementary Fig. S2d).

And on p. 21 as follows:

The J2 antibody has been reported to underestimate dsRNA localization²⁶; however, using optimized antibody concentrations (Supplementary Fig. S19, S20) and optimized staining protocols as detailed in Methods, we achieved excellent sensitivity to dsRNA with signal present in all infected cells, even in early infection with very low vgRNA levels (Supplementary Fig. S2d).

2) I find it quite strange that overall nsp12 numbers don't increase between 6 and 24 hours post infection. One possibility that I would like to see addressed if possible is that the nsp12 antibody used in the study isn't actually binding to the actively replicating form of the protein (which would in fact be part of a large complex of nsps and so assumed to be quite inaccessible). If this is the case it may explain why there isn't a large increase in nsp12 molecules detected.

We agree that it is useful to provide evidence that nsp12 immunofluorescence imaging highlights replication competent complexes. To do this, we have also imaged two additional RdRp subunits, nsp7 and nsp8, and found both of them colocalizing with the vgRNA clusters. We further imaged the nsp12 and nsp8 pair, again finding strong colocalization. The text on p. 13 has been modified as follows:

To verify that nsp12 labeling is a good reporter of assembled replication complexes, we have also imaged two accessory subunits of RdRp, nsp7 and nsp8. We find close association of these subunits with vgRNA as shown in Fig. 3e, f, and in the pair-correlation functions of Fig. 3i (see also Supplementary Fig. S4 and S5). Nsp12 and nsp8 colocalized with each other on the nanoscale (Supplementary Fig. S6), indicating their interaction within ROs, as expected for subunits of assembled RdRp.

Moreover, we also found the association of newly synthesized RNA labelled by BrU with RdRp, please see new Supplementary Fig. S8. In this regard, the text on p. 13 has been modified as follows:

Confocal and SR imaging localizes RNA-containing BrU to the perinuclear clusters of vgRNA (Fig. 3g, Supplementary Fig. S7) and close to nsp12 (Supplementary Fig. S8), further proving that these structures are the sites of active replication and transcription of viral RNA.

Minor:

1) It may just be a personal preference, but I found it a little hard to follow the order of the figures, particularly where the graphs are contained within other images. Whilst it may not be the most efficient use of space it may aid the reader to have the various panels presented in the order that they are described in the text, and with the graphs placed separately alongside the images.

We thank the referee for this helpful suggestion. We have now changed the order of the panels and have placed the insets outside the cell images wherever possible.

2) Fig S1 J, is vgRNA also visible inside spike labelled virions at the cell periphery?

Even though the labelling efficiency of vgRNA inside assembled virions is low due to the tight packaging of vgRNA, we do observe vgRNA inside selected virions released from infected cells. Examples of these are now displayed in the new Supplementary Fig. S17b. The manuscript has been updated on p. 18 as follows:

Spike labeling forms typical ~ 150 nm hollow particles at the cell periphery, and we detect weak vgRNA signal in the center of some of these particles (Supplementary Fig. S17b), consistent with the structure of SARS-CoV-2 virions that contain a single vgRNA molecule.

3) Page 7, line 148. Can you speculate why the observed circular structures that the vgRNA forms may be hollow? Might this support a model whereby vgRNA is copied by polymerase located at the DMV membrane, i.e. is this suggesting that the viral polymerase may be membrane-associated?

While this is a tantalizing suggestion, we observe RdRp complexes and BrU-RNA located tightly around the vgRNA clusters, but also inside them. Our data does not therefore provide evidence that the replication occurs only at the peripheral DMV membrane. We do, however observe dsRNA preferentially in the central voids of vgRNA structures (described on p. 9) which suggests that dsRNA accumulates within separate regions of ROs. Detailed spatial regulation of replication within ROs is an important topic indeed and may be a subject of future studies.

4) Figure S3 A, please could you indicate on the images some examples of which clusters are taken to be single vgRNA punctum (those that are used as calibration)?

We have now pointed out clusters of vgRNA localizations with circles (See Supplementary Fig. S3a) and added examples of these low-density areas in Fig. 1e,i. The algorithm used to find the number of

molecules per punctum is described in the Methods, and we have now specifically mentioned this on p8:

(Supplementary Fig. S3b-c; procedure detailed in Methods).

5) Page 11, line 220. Fig 3g is referenced in the text before e and f, which is hard to follow.

Following the reviewer's suggestion, we have renumbered and rearranged panels in Fig. 3 with the addition of new panels 3e,f,g,i.

6) Page 12, Figure 3h seems a little out of place. It isn't mentioned in the text until much later, making it difficult to understand the relevance of this panel in figure 3, and so I would suggest moving this panel to Figure 4.

We thank the referee for this suggestion, Figure 3h indeed appeared somewhat out of place. We have now placed the data from old Figure 3h within Figure 4b.

REVIEWERS' COMMENTS

Reviewer #1 (Remarks to the Author):

I believe the authors have largely addressed my major concerns of the original submission.

This study should provide new mechanistic insight into SARS-CoV-2 replication and I support its publication.

Reviewer #2 (Remarks to the Author):

The authors have adequately addressed all of my concerns and I'm happy to recommend the manuscript for publication as it now stands.